# Budgeted-Bandits with Controlled Restarts with Applications in Learning and Computing

**Semih Cayci**[*]                                                                *cayci@mathc.rwth-aachen.de*
*Department of Mathematics*
*RWTH Aachen University*

**Yilin Zheng**[*]                                                                *yilinzheng@google.com*
*Google*

**Atilla Eryilmaz**                                                                *eryilmaz.2@osu.edu*
*Department of Electrical and Computer Engineering*
*The Ohio State University*

**Reviewed on OpenReview:** *https://openreview.net/forum?id=lvb5qDAa4B*

## Abstract

Maximizing the cumulative reward of a sequence of tasks under a time budget has been ubiquitous in many applications in computing and machine learning. Often, tasks can have random completion time and the controller needs to learn the unknown statistics while making optimal decisions. In addition to the classic exploration-exploitation trade-off, it has been shown that restarting strategy can boost the performance of the control algorithm by interrupting ongoing tasks at the expense of losing its reward. In this work, we consider a bandit setting where each decision takes a random completion time and yields a random (possibly correlated) reward at the end, both with unknown values at the time of decision. The goal of the decision-maker is to maximize the expected total reward subject to a stringent time budget $\tau$. As an additional control, we allow the decision-maker to interrupt an ongoing task and forgo its reward for a potentially more rewarding restart. Unlike previous works, we do not: assume any prior knowledge on the system statistics, or limit the action space of restarting strategies to be finite. Under this general framework, we developed efficient bandit algorithms to find optimal arms and restart strategies with $O(\log(\tau))$ and $O(\sqrt{\tau \log(\tau)})$ regret for both finite and continuous set of restart times, respectively. Furthermore, through numerical studies, we verified the applicability of our algorithm in the diverse contexts of: (i) algorithm portfolios for SAT solvers; (ii) task scheduling in wireless networks; and (iii) hyperparameter tuning in neural network training.

## 1 Introduction

Processes under a time budget, which continue until the total time spent exceeds a given time horizon, have long been a focal point of scientific interest as a consequence of their universal applicability in a broad class of disciplines in computing and machine learning (Redner, 2001; Condamin et al., 2007). It has been shown that any process with time budget can employ controlled restarts with the goal of expediting the completion times, thus increasing the time-efficiency of stochastic systems (Pal & Reuveni, 2017). Consequently, restart strategies have attracted significant attention to boost the time-efficiency of stochastic systems in various contexts. They have been extensively used to study diffusion mechanics (Evans & Majumdar, 2011; Pal et al., 2016), target search applications (Kusmierz et al., 2014; Eliazar et al., 2007), throughput maximization (Asmussen et al., 2008), and run-times of randomized algorithms (Hoos & Stützle, 2004; Luby et al., 1993;

---

[*]Equal contribution

Selman et al., 1994). In particular, they have been widely used as a tool for the optimization of randomized algorithms that employ stochastic local search methods whose running times exhibit heavy-tailed behavior (Luby et al., 1993; Gomes et al., 1998).

In this paper, we investigate the cumulative reward maximization problem in the context of budgeted decision processes under controlled restarts in continuous time. Building on the knapsack bandit framework, we address the challenge of selecting the optimal restart time in addition to the exploration-exploitation trade-off. This learning problem has unique dynamics: the cumulative reward function is a controlled and stopped random walk with potentially heavy-tailed increments, and the restart mechanism leads to right-censored feedback, which imposes a specific information structure. In order to design efficient learning algorithms that fully capture these dynamics, we incorporate new design and analysis techniques from bandit theory, renewal theory, and statistical estimation. For both finite and continuous sets of restart times, we develop provably efficient online learning algorithms to learn the optimal action and restart strategy at the same time.

In addition to the theoretical performance guarantee, we investigate the budgeted-bandits learning problem under controlled restart strategies through numerical studies. In particular, our empirical studies show that our algorithm can be useful in boosting the performance of SAT solvers, scheduling tasks in distributed networks, and deciding the early stopping time when training neural networks. These results verify the applicability of this general framework and the performance of the proposed algorithms.

## 1.1 Related Work

Bandit problem with knapsack constraints and its variants have been studied extensively in the last decade. In György et al. (2007); Badanidiyuru et al. (2013); Combes et al. (2015); Xia et al. (2015); Tran-Thanh et al. (2012); Cayci et al. (2020), the problem was considered in a stochastic setting. This basic setting was extended to linear contextual setting (Agrawal & Devanur, 2016), combinatorial semi-bandit setting (Sankararaman & Slivkins, 2017), adversarial bandit setting (Immorlica et al., 2019). For further details in this branch of bandit theory, we refer to Slivkins et al. (2019). Bandits with delayed feedback (Joulani et al., 2013; Vernade et al., 2020; Cesa-Bianchi et al., 2016) allow the reward to be delayed. However, as a substantial difference, these works do not incorporate a restart or cancellation mechanism into the learning problem. While in our work, restarting improves reward-per-unit-time due to increasing residual completion times.

The restart mechanisms have been popular, particularly in the context of boosting the Las Vegas algorithms. The pioneering work in this branch is Luby et al. (1993), where a minimax restart strategy is proposed to achieve optimal expected run-time up to a logarithmic factor. In Gagliolo & Schmidhuber (2007), a hybrid learning strategy between Luby scheme and a fixed restart strategy is developed in the adversarial setting. In Streeter & Golovin (2009), an algorithm portfolio design methodology that employs a restart strategy is developed as an extension of the Exp3 Algorithm in the non-stochastic bandit setting, and polynomial regret is shown with respect to a constant factor approximation of the optimal strategy. These works are designed in a non-stochastic setting for worst-case performance metrics. In contrast, our work incorporates the statistical structure of the problem and learns the underlying statistics of the system based on empirical observations. Note that this restarting mechanism is completely different from the stochastic bandit (Cheung et al., 2019; Zhao et al., 2020; Kang et al., 2023) setup where restarting is for the sampling process, for example, using a sliding window or discounted factor to forget obsolete information.

Combining the budgeted-bandits setting and controlled restart strategy is a particularly challenging problem. In Cayci et al. (2019), they tried to address the problem with prior knowledge of the moments. In addition, they only considered uncorrelated rewards and a finite set of restart times. Later works like Liu & Fang (2023; 2024) extended the framework by adding additional stochastic constraints. However, they still focused on a limited set of restart times. Our work improves on previous works by eliminating the necessity of prior statistical knowledge and allowing a general (possibly continuous) set of restart times. In addition, we can also allow correlations between the completion time and reward.

## 1.2 Contributions

In light of the related work above , our main contributions in this paper are as follows:

- This work provides a principled approach to continuous-time exploration-exploitation problems that require restart strategies for optimal time efficiency. We study and explain the impact of restart strategies in a general knapsack bandit setting that includes potentially heavy-tailed and *correlated* time-reward pairs for each arm. The design and regret analysis of learning algorithms require tools from renewal theory and stochastic control, as well as concentration inequalities for rate estimation. These results can be useful in other bandit problems as well (cf. Section 2 to Section 4).

- For a finite set of restart times, we study algorithms that use variance estimates and the information structure that stems from the right-censored feedback so as to achieve $O(\log(\tau))$ regret bounds with no prior knowledge. In particular, we propose:
    - the UCB-RB Algorithm based on rate estimation based on sample-mean estimator to achieve particularly good performance for finite and small restart times (cf. Theorem 2).
    - the UCB-RM Algorithm based on median-boosted rate estimation *without* assuming any knowledge of higher-order moments to achieve good performance in a very general setting of large (potentially infinite) restart times at the expense of degraded performance for small restart times (cf. Theorem 1).

- For a continuous decision set for restart strategies, we propose an algorithm called UCB-RC that achieves $O(\sqrt{\tau \log(\tau)})$ regret (cf. Theorem 3).

- We evaluate the performance of the learning algorithm developed in this paper for boosting the performance of SAT solver, task scheduling in distributed networks, and finding early stopping time for training neural networks (cf. Section 6).

## 1.3 Notation

In this subsection, we define some notation that will be used throughout the paper. $\mathbb{I}$ denotes the indicator function. For any $k \in \mathbb{Z}_+$, $[k]$ denotes the set of integers from 1 to $k$, i.e., $[k] = \{1, 2, \ldots, k\}$. For $x \in \mathbb{R}$, $(x)_+ = \max\{0, x\}$. For two real numbers $x, y$, we define $x \wedge y = \min\{x, y\}$ and $x \vee y = \max\{x, y\}$.

## 1.4 Motivating Examples

The problem we proposed has many potential real-world applications where a controller has to make decisions (possibly with restart) under a time budget with no prior knowledge of the statistics. In this section, we provide some motivating examples that fall within the scope of this framework.

*(Algorithm Portfolio Selection for SAT Solvers)* Boolean satisfiability (SAT) is a canonical NP-complete problem (Hoos & Stützle, 2004; Arora & Barak, 2009). As a consequence, the design of efficient SAT solvers has been an important long standing challenge, and an annual SAT Competition is held for SAT solvers (Heule et al., 2019). In these competitions, the objective of a SAT solver is to solve as many problem instances as possible within a given time interval $[0, \tau]$, which defines a budgeted decision process. As a result of the completion time distributions, the restart strategies have been an essential part of the SAT solvers (Luby et al., 1993; Selman et al., 1994). Therefore, the proposed framework can be used as a meta-learning algorithm to improve the performance of SAT solvers.

*(Task Scheduling in Distributed Server System)* In many distributed server systems such as web servers, data base servers, and computing clusters, tasks for service arrive randomly and must be assigned to one of the host machines for processing (Liu & Fang, 2023). This is especially important if the communication is through wireless channels. Under a limited time interval $[0, \tau]$, a controller will try to maximize the number of completed tasks. It has been shown that the task completion time has a heavy-tail distribution in many applications (Harchol-Balter, 1999). Therefore, incorporating a restarting strategy as designed in the proposed framework can be useful.

*(Tuning the Stopping Time for Neural Network Training)* It is well-known that over-fitting can happen when training a machine learning model such as a neural network on a training set for too many steps (Caruana et al., 2000; Prechelt, 2002). Early stopping is a common strategy to terminate the training in order to achieve higher testing performance on unseen data. It has been shown that early-stopped neural networks are consistent (Ji et al., 2021) and more robust to noise (Li et al., 2020). However, the decision of early stopping is often made based on experience of specific networks and applications. As an alternative, our framework can be used as a general control method to boost the testing performance while finding an optimal early stopping time.

We will revisit these examples with simulations in Section 6. In the following section, we describe the problem setting in detail.

## 2 System Setup

We consider a budgeted decision process with a given time horizon $\tau > 0$. The increments of the process are controlled as follows: if arm $k \in [K]$ is selected at the $n$-th trial, the random completion time is $X_{k,n}$, that is independent across $k$ and independent over $n$ for each $k$ with the following weak conditions:

$$\mathbb{E}[X_{k,1}^p] < \infty, \ \exists p \geq 4, \tag{1}$$

$$X_{k,n} > 0, \ a.s. \tag{2}$$

Note that $X_{k,n}$ can potentially be a heavy-tailed random variable. We do not assume any prior knowledge of the higher-order statistics, and the bounded fourth-order moment assumption enables us to use empirical Bernstein inequality, which operates without any prior statistical knowledge. Pulling arm $k$ at trial $n$ yields a reward $R_{k,n}$ at the end. The joint variable $(X_{k,n}, R_{k,n})$ is sampled identically for every $n$. Note that we allow $R_{k,n}$ to be possibly correlated with $X_{k,n}$, and for simplicity, we assume that $R_{k,n} \in [0,1]$ almost surely.

In our framework, the controller has the option to interrupt an ongoing task and restart it with a potentially different arm. Namely, for a set of *restart* (or *cutoff*) times $\mathbb{T} \subset [0, \infty]$, the controller may prefer to restart the task from its initial after time $t \in \mathbb{T}$ at the loss of the ultimate rewards and possible additional time-varying cost of restarting the process. We model the resetting time (cost of restarting) as $C_k(t) \in [0, t]$, which is a deterministic function of $t \in \mathbb{T}$ known by the controller for simplicity. Note that the design and analysis presented in this paper can be extended to random resetting time processes (Evans & Majumdar, 2011).

If the $n$-th trial of arm $k$ is restarted at time $t \in \mathbb{T}$, then the resulting total time and reward are as follows:

$$U_{k,n}(t) = \min\{X_{k,n}, t\} + C_k(t)\mathbb{I}\{X_{k,n} > t\}, \tag{3}$$

$$V_{k,n}(t) = R_{k,n}\mathbb{I}\{X_{k,n} \leq t\}, \tag{4}$$

Note that the feedback $(U_{k,n}(t), V_{k,n}(t))$ in this case is right-censored and the time-cost here evolves in continuous time.

Incorporating the restart mechanism, the control of the decision-maker consists of two decisions at $n$-th trial:

$$\pi_n = (I_n, \nu_n) \in [K] \times \mathbb{T}.$$

Here, $I_n \in [K]$ denotes the arm decision, and $\nu_n \in \mathbb{T}$ denotes the restart time decision. Under a policy $\pi = \{\pi_n : n \geq 1\}$, let

$$\mathcal{F}_n^\pi = \sigma\left(\{(U_{I_i,i}(\nu_i), V_{I_i,i}(\nu_i), \mathbb{I}\{X_{I_i,i} \leq \nu_i\} : i = 1, 2, \ldots, n\}\right),$$

be the history until $n$-th trial. We call a policy $\pi$ admissible if $\pi_{n+1}$ is $\mathcal{F}_n^\pi$-measurable for all $n \geq 1$.

For a given time horizon $\tau > 0$, the objective of the decision-maker is to maximize the expected cumulative reward collected in the time interval $[0, \tau]$. Specifically, letting

$$N_\pi(\tau) = \inf\left\{n : \sum_{i=1}^n U_{I_i,i}(\nu_i) > \tau\right\}, \tag{5}$$

be the number of pulls under policy $\pi$. The number of pulls depends on $\pi$ since each decision incurs a *decision-dependent* random time, which is subject to a stringent budget. The cumulative reward under $\pi$ is defined as follows:

$$\texttt{REW}_\pi(\tau) = \sum_{n=1}^{N_\pi(\tau)} V_{I_n,n}(\nu_n). \tag{6}$$

The decision-maker attempts to achieve the optimal reward:

$$\texttt{OPT}(\tau) = \max_{\pi \in \Pi} \ \mathbb{E}[\texttt{REW}_\pi(\tau)], \tag{7}$$

where $\Pi$ is the set of all admissible policies. Equivalently, it aims to minimize the expected regret:

$$\texttt{REG}_\pi(\tau) = \texttt{OPT}(\tau) - \mathbb{E}[\texttt{REW}_\pi(\tau)]. \tag{8}$$

## 2.1 Right-Censored Feedback and Information Structure

Since the feedback we obtain for a decision of $(k, t)$ is the pair of right-censored random variables $(U_{k,n}(t), V_{k,n}(t))$ as in equation 3, for any $t' \leq t$, we have the following:

$$U_{k,n}(t') \text{ is } \sigma\big(U_{k,n}(t)\big)\text{-measurable},$$
$$V_{k,n}(t') \text{ is } \sigma\big(V_{k,n}(t)\big)\text{-measurable}.$$

In other words, the feedback from a restart time decision $t > 0$ can be faithfully used as feedback for another restart time decision $t' \leq t$. This implies the information structure is asymmetric, i.e. the information gain by a large $t \in \mathbb{T}$ is larger compared to $t' \leq t$. As we will see later, learning effectively from the right-censored data using the information structure it incurred is important for designing an effective online learning algorithm.

## 3 Summary of Main Results

In this section, we present the results of the paper informally.

**Proposition 1** (Informal Regret Upper Bound for UCB-RM)**.** *For a finite set of restarting times $|\mathbb{T}| = L$ with $\infty \in \mathbb{T}$ and $K$ arms, the regret of UCB-RM is*

$$\texttt{REG}_{\pi^{\texttt{M}}}(\tau) \leq \alpha \log(\tau) \sum_k c_k^{\texttt{M}} + O(KL),$$

*where $c_k^{\texttt{M}} > 0$, $k \in [K]$ are constants (with respect to $\tau$) that depend on the higher-order moments of the random completion time and reward for arm $k$. The full statement with exact constants is given in Theorem 1.*

**Proposition 2** (Informal Regret Upper Bound for UCB-RB)**.** *For a finite set of restarting times $|\mathbb{T}| = L$ with $\infty \notin \mathbb{T}$ and $K$ arms, the regret of UCB-RB is*

$$\texttt{REG}_{\pi^{\texttt{B}}}(\tau) \leq \alpha \log(\tau) \sum_k c_k^{\texttt{B}} + O(KL),$$

*where $c_k^{\texttt{B}} > 0$, $k \in [K]$ are constants (with respect to $\tau$) that depend on the higher-order moments of the random completion time and reward for arm $k$. Typically, these coefficients are smaller than $c_k^{\texttt{M}}$. The full statement with exact constants is given in Theorem 2.*

Note that in the general definition of $\mathbb{T}$, we included the action $\infty$ in $\mathbb{T}$, which corresponds to the case where each arm pull is completed without restarting. In Proposition 2, we impose the condition that each arm has a finite restart time to obtain the regret guarantees for UCB-RB.

**Proposition 3** (Informal Regret Upper Bound for UCB-RC)**.** *Using a fixed discretization step-size for the continuous set of restarting time, the regret of UCB-RC satisfies*

$$\limsup_{\tau \to \infty} \ \frac{\texttt{REG}_{\pi^{\texttt{C}}}(\tau)}{\sqrt{\tau \log(\tau)}} \leq c. \tag{9}$$

where $c$ is a problem dependent constant that is independent of time $\tau$. The full statement with exact constants is given in Theorem 3. In the next section, we first give an asymptotically-optimal offline policy as a baseline.

## 4 Asymptotically-Optimal Offline Policy with Known Statistics

The control problem described in equation 7 is a variant of the well-known unknown stochastic knapsack problem (Kleinberg & Tardos, 2006). In the literature, there are similar problems that are known to be PSPACE-hard (Papadimitriou & Tsitsiklis, 1999; Badanidiyuru et al., 2013). Therefore, we need to first find a tractable algorithm to approximate the optimal policy. In this section, we propose a simple policy for the problem introduced in Section 2, and prove its efficiency by using the theory of renewal processes and stopping times.

First, we define the (renewal) reward rate, which will be the main quantity of interest when designing algorithms.

**Definition 1** (Reward Rate). *For a decision $(k, t)$, the (renewal) reward rate is defined as follows:*

$$r_k(t) = \frac{\mathbb{E}[R_{k,1}\mathbb{I}\{X_{k,1} \leq t\}]}{\mathbb{E}[(X_{k,1} \wedge t) + C_k(t)\mathbb{I}\{X_{k,1} > t\}]} = \frac{\mathbb{E}[V_{k,1}(t)]}{\mathbb{E}[U_{k,1}(t)]}. \tag{10}$$

$r_k(t)$ is the growth rate of the expected total reward over time if the controller persistently chooses the action $(k, t)$, known as the reward rate in renewal theory. In other words, the reward rate represents the time average reward per unit time. As a consequence of the elementary renewal theorem (Gut, 2009), the reward of the offline policy that persistently makes a decision $(k, t)$ is $\mathbb{E}[\texttt{REW}_\pi(\tau)] = r_k(t) \cdot \tau + o(\tau)$. In the following, we will provide an upper bound for $\texttt{OPT}(\tau)$ based on $r_k(t)$ and the time horizon $\tau$.

**Proposition 4** (Upper bound for `OPT`). *Let the optimal reward rate be defined as follows:*

$$r^* = \arg\sup_{(k,t)} r_k(t). \tag{11}$$

*If there exists a $p_0 > 2$ and $u < \infty$ such that $\mathbb{E}[(X_{k,1})^{p_0}] \leq u$ holds for all $k \in [K]$, then we have the following upper bound for `OPT`:*

$$\texttt{OPT}(\tau) \leq r^*\big(\tau + \Phi(u)\big), \tag{12}$$

*for any $\tau > 0$ where $\Phi(u)$ is a constant that is independent of $\tau$.*

*Proof.* The proof generalizes the optimality gap results in (Xia et al., 2015; Cayci et al., 2020). Under any admissible policy $\pi \in \Pi$, an extension of Wald's identity yields the following upper bound:

$$\mathbb{E}[\texttt{REW}_\pi(\tau)] \leq r^*\mathbb{E}[U_{N_\pi(\tau)}],$$

where $N_\pi(\tau)$ is the first-passage time of the controlled random walk under $\pi$. The excess over the boundary, $\mathbb{E}[U_{N_\pi(\tau)}] - \tau$, is known to be $o(\tau)$ for simple random walks by the elementary renewal theorem (Gut, 2009). For controlled random walks, (Lalley & Lorden, 1986) shows that $\mathbb{E}[U_{N_\pi(\tau)}] - \tau = O(u)$, which is independent of $\tau$, if $\mathbb{E}[X_{k,1}^{p_0}] \leq u < \infty$ holds for all $k \in [K]$ for some $p_0 > 2$. Thus, under the moment assumption equation 1, we have $\mathbb{E}[U_{N_\pi(\tau)}] - \tau = O(1)$ as $\tau \to \infty$. $\qquad\square$

From the discussion above, it is natural to consider an algorithm that optimizes $r_k(t)$ among all decisions $(k, t)$ as an approximation of the optimal policy. Accordingly, the optimal offline policy, denoted as $\pi^{\texttt{off}}$, is given in Algorithm 1.

The performance analysis of $\pi^{\texttt{off}}$ is fairly straightforward since the random process it induces is a simple random walk.

---

**Algorithm 1:** Asymptotically-Optimal Offline Policy $\pi^{\texttt{off}}$

---

$n = 0$;
$S_n = 0$;
**while** $S_n \leq \tau$ **do**
    $(k^*, t^*) = \arg\sup\limits_{(k,t)} \ r_k(t)$;                     // Maximize equation 10
    **if** $X_{k^*,n} \leq t^*$ **then**
        $S_{n+1} = S_n + X_{k^*,n}$;
        Obtain reward $R_{k^*,n}$;
    **else**
        $S_{n+1} = S_n + t^* + C_{k^*}(t^*)$;                 //Restart time
    $n = n + 1$;                             //New epoch starts

---

**Proposition 5.** *The reward under $\pi^{\texttt{off}}$ is bounded as follows:*

$$r^* \tau \leq \mathbb{E}[\texttt{REW}_{\pi^{\texttt{off}}}(\tau)] \leq r^* \left( \tau + \frac{\mathbb{E}[Y_*^2]}{\left( \mathbb{E}[Y_*] \right)^2} \right), \tag{13}$$

*where $Y_* = U_{k^*,1}(t^*)$ is the completion time of an epoch under $\pi^{\texttt{off}}$, and $(k^*, t^*)$ is defined in Algorithm 1.*

The proof of Proposition 5 follows from Lorden's inequality for renewal processes (Asmussen, 2008).

As a consequence of Proposition 4 and Proposition 5, the optimality gap of the offline policy is bounded for all $\tau > 0$.

**Corollary 1** (Optimality Gap of $\pi^{\texttt{off}}$). *For any $\tau > 0$, the optimality gap of $\pi^{\texttt{off}}$ is bounded:*

$$\texttt{OPT}(\tau) - \mathbb{E}[\texttt{REW}_{\pi^{\texttt{off}}}(\tau)] \leq r^* \Phi(u), \tag{14}$$

*where $\Phi(u)$ is the constant in Proposition 4.*

As we can see, the expected reward of the simple offline policy only has a bounded gap with the best possible policy, which can only be achieved by an NP-hard algorithm. In addition, we observed that the reward rate $r_k(t)$ is the dominant component of the cumulative reward. Next, we will analyze the behavior of $r_k(t)$ with respect to the restart time $t$.

## 4.1 When is it optimal to restart?

For any given arm $k \in [K]$ and any set of restart times $\mathbb{T}$, let the optimal restart time be defined as follows:

$$t_k^* \in \arg\sup\limits_{t \in \mathbb{T}} \ r_k(t). \tag{15}$$

If the completion time $X_{k,n}$ and reward $R_{k,n}$ are independent, then it was shown in (Cayci et al., 2019) that it is optimal to restart a cycle at a finite time if the following condition holds:

$$\mathbb{E}[X_{k,1} - t | X_{k,1} > t] > \mathbb{E}[X_{k,1}] \tag{16}$$

for $t \in \mathbb{T} \setminus \{\infty\}$, and it is noted that all heavy-tailed and some light-tailed completion time distributions satisfy this condition. Exponential distribution corresponds to the boundary case where $\mathbb{E}[X_{k,1} - t | X_{k,1} > t] = \mathbb{E}[X_{k,1}]$ for all $t \geq 0$, which follows from the memoryless property. If $X_{k,1}$ and $R_{k,1}$ are correlated, this is no longer true and the situation is more complicated. In the following, we extend this result to correlated $(X_{k,1}, R_{k,1})$ pairs.

**Proposition 6** (Optimal Restart Time). *For a given arm $\left( X_{k,1}, R_{k,1}, C_k(t) \right)$, we have $t_k^* < \infty$ if and only if the following holds:*

$$\frac{\mathbb{E}\left[R_{k,1} | X_{k,1} > t\right]}{\mathbb{E}\left[X_{k,1} - \left( t + C_k(t) \right) | X_{k,1} > t\right]} < \frac{\mathbb{E}[R_{k,1}]}{\mathbb{E}[X_{k,1}]}, \tag{17}$$

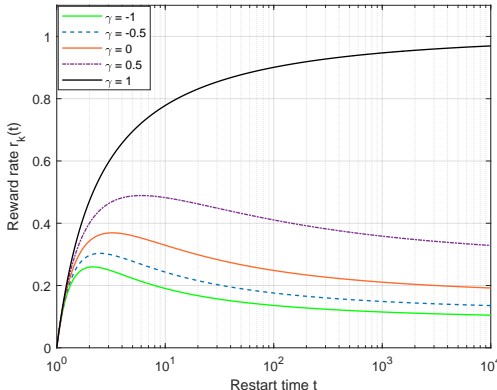

Figure 1: Impact of correlation between $X_{k,n}$ and $R_{k,n}$ on the optimal restart time for $X_{k,n} \sim Pareto(1, 1.2)$ and $\gamma \in [-1, 1]$. Positive correlation between the completion time and reward leads to higher restart times, and waiting until the completion of every task is optimal for $\gamma \geq 1$ since waiting becomes more rewarding.

*for some $t > 0$.*

The proof follows from showing equation 17 is equivalent to $r_k(t) > \mathbb{E}[R_{k,1}]/\mathbb{E}[X_{k,1}]$ for any $t > 0$.

Interpretation of Proposition 6 is as follows: for any restart time $t \in \mathbb{T} \setminus \{\infty\}$, if the reward rate of waiting until the completion (i.e., *the residual reward rate*) is lower than the reward rate of a new trial, then it is optimal to restart.

Note that equation 16 is a special case of Proposition 6 with immediate returns, i.e., $C_k(t) = 0$, and $X_{k,n}$ and $R_{k,n}$ are independent.

### 4.2 What is the impact of correlation on the restart time?

The correlation between $X_{k,n}$ and $R_{k,n}$ has a substantial impact on whether the optimal restart times are finite or not. As an example, let $X_{k,n} \sim Pareto(1, \alpha)$ for some $\alpha \in (1, 2)$, and $C_k(t) = 0$ for all $t$.

1. If $X_{k,n}$ and $R_{k,n}$ are independent, then equation 16 holds and $t_k^* < \infty$, i.e., it is optimal to restart after a finite time.

2. If $R_{k,n} = \omega X_{k,n}^\gamma$ for some $\omega > 0$ and $\gamma \geq 1$, then it is optimal to wait until the end of the task, i.e., $t_k^* = \infty$.

3. If $R_{k,n} = \omega X_{k,n}^\gamma$ for $\omega > 0$ and $\gamma < 1$, then we have $t_k^* < \infty$.

The impact of the correlation between $X_{k,n}$ and $R_{k,n}$ on the behavior of optimal restart time is illustrated in Figure 1.

In the next section, we develop online learning algorithms for the problem, and present the regret bounds.

## 5 Online Policy for Controlled Restarts with Unknown Statistics

In this section, we develop online learning algorithms with provably good performance guarantees. The right-censored nature of the feedback due to the restart mechanism imposes an interesting information structure to this problem. We first describe the nature of this information structure.

We consider a finite $\mathbb{T} = \{t_1, t_2, \ldots, t_L\}$ such that

$$t_1 < t_2 < \ldots < t_L \leq \infty. \tag{18}$$

Throughout the paper, we assume that any action set $\mathbb{T}$ satisfies the following assumption:

**Assumption 1.** *Given a decision set $\mathbb{T}$, there exists $\epsilon, \mu_* > 0$ that satisfies the following:*

$$\mathbb{E}[\min\{X_{k,1}, t_1\}] \geq \mu_*,$$

*and*

$$\mathbb{P}(X_{k,1} \leq t_1 | R_{k,1} = \rho) \geq \epsilon, \ \forall \rho \in [0,1],$$

*for all $k \in [K]$, where $t_1 = \min \mathbb{T}$.*

Note that Assumption 1 is a simple technical condition that ensures efficient estimation of $r_k(t)$ from the samples of $(X_{k,n}, R_{k,n})$ for all $t \in \mathbb{T}$.

In order to capture the benefits of the information structure, for arm $k$ and restart time decision $t_l$, let

$$\mathcal{I}_{k,l}(n) = \{i \leq n : \pi_i = (k, t_l)\}.$$

Then, the available feedback for a decision $(k, t_l)$ is as follows:

$$\tilde{\mathcal{I}}_{k,l}(n) = \bigcup_{l' \geq l} \mathcal{I}_{k,l}(n).$$

The size of $\tilde{\mathcal{I}}_{k,l}(n)$, i.e., the number of samples available for $(k, l)$ is defined as follows:

$$\tilde{T}_{k,l}(n) = |\tilde{\mathcal{I}}_{k,l}(n)| = \sum_{l' \geq l} T_{k,l'}(n), \tag{19}$$

From above, it is observed that the information structure increases the number of samples substantially for each decision, i.e., $\tilde{T}_{k,l}(n) \geq T_{k,l}(n)$ for all $k, l$.

Compared with previous works (Cayci et al., 2019; Liu & Fang, 2023; 2024), we allow a general set of restarting times. Under this general framework, the radius of the set of restart times $\mathbb{T}$ has a crucial impact on algorithm design and performance, depending on the tail distributions of the completion times. In the following subsection, we propose two algorithms for both small and large (potentially infinite) $t_L$, and compare their characteristics.

### 5.1 Finite Set of Restart Times: UCB-RM and UCB-RB

In this section, we considered a finite set of restart times and propose two algorithms UCB-RM and UCB-RB based on median-of-mean estimators and empirical mean estimators respectively (Algorithm 2). Note that the information structure that stems from the right-censored feedback is utilized. As we will see, this information structure will lead to substantial improvements in performance.

#### 5.1.1 UCB-RM Algorithm

For an index set $S \subset \mathbb{Z}_+$, let the empirical mean $\widehat{\mathbb{E}}_S$ and empirical variance $\mathbb{V}_S$ of a random sequence $\{Y_i : i \geq 1\}$ be defined as follows:

$$\widehat{\mathbb{E}}_S[Y] = \frac{1}{|S|} \sum_{i \in S} Y_i, \tag{20}$$

$$\mathbb{V}_S[Y] = \frac{1}{|S|} \sum_{i \in S} \left(Y_i - \widehat{\mathbb{E}}_S[Y]\right)^2. \tag{21}$$

In general, if the completion time distributions are such that the optimal restart time is very large or potentially infinite for some arms, the radius of the set of restarting time $\mathbb{T}$ will be large. In this case, the empirical rate estimation will not provide fast enough convergence rates, which can lead to substantially deteriorated regret bounds.

In order to overcome this, we design a UCB-type policy that incorporates a median-based rate estimator which is defined in Definition 2. This estimator can achieve good performance in a very general setting that allows not restarting as a possible action, i.e., we consider a decision set

$$\mathbb{T} = \{t_1 < t_2 < \ldots < t_L = \infty\},$$

where $t_L = \infty$ implies the controller can wait until the task is completed.

**Definition 2** (Median-based rate estimator). *Consider $(k, t_l) \in [K] \times \mathbb{T}$, and let $G_1, G_2, \ldots, G_m$ be a partition of the set $\tilde{\mathcal{I}}_{k,l}(n)$ such that $|G_j| = \lfloor \tilde{T}_{k,l}(n)/m \rfloor$, $\forall j \in [m]$ for $m = \lfloor 3.5\alpha \log(n) \rfloor + 1$, $\alpha > 2$. The median-of-means estimator for $U_{k,n}(t_l)$ is defined as follows:*

$$\mathbb{M}_{\tilde{\mathcal{I}}_{k,l}(n)}(U_k(t_l)) = \texttt{median}\Big\{\widehat{\mathbb{E}}_{G_1}[U_k(t_l)], \widehat{\mathbb{E}}_{G_2}[U_k(t_l)], \ldots, \widehat{\mathbb{E}}_{G_m}[U_k(t_l)]\Big\}, \tag{22}$$

*where $\widehat{\mathbb{E}}_S[X]$ is the empirical mean of a sequence $X$ over the index set $S$ defined in equation 20. Then, the median-based rate estimator for the action $(k, t_l)$ is defined as follows:*

$$\widehat{r}_{k,l,n}^{\texttt{M}} = \frac{\mathbb{M}_{\tilde{\mathcal{I}}_{k,l}(n)}(V_k(t_l))}{\mathbb{M}_{\tilde{\mathcal{I}}_{k,l}(n)}(U_k(t_l))}. \tag{23}$$

Intuitively, the median-of-mean estimators are more robust to potentially large samples caused by heavy-tail distribution.

Similarly, for variance estimation, using the empirical variance $\mathbb{V}_S$ defined in equation 21, we can define

$$\mathbb{V}_{\tilde{\mathcal{I}}_{k,l}(n)}^{\texttt{M}}(U_k(t_l)) = \texttt{median}\Big\{\mathbb{V}_{G_1}[U_k(t_l)], \ldots, \mathbb{V}_{G_m}[U_k(t_l)]\Big\},$$

With these median-based estimators, we can construct the confidence radii as

$$c_{k,l,n}^{\texttt{M}} = \frac{(1+\beta)^2}{(1-\beta)} \cdot \frac{\eta_{k,l,n}^{\texttt{M}} + \widehat{r}_{k,l,n}^{\texttt{M}} \cdot \epsilon_{k,l,n}^{\texttt{M}}}{\mathbb{M}_{\tilde{\mathcal{I}}_{k,l}(n)}(U_k(t_l))}, \ \beta \in (0, 1), \tag{24}$$

where the parameters are defined as

$$\epsilon_{k,l,n}^{\texttt{M}} = 11\sqrt{\frac{2\mathbb{V}_{\tilde{\mathcal{I}}_{k,l}(n)}^{\texttt{M}}[U_k(t_l)] \log(n^\alpha)}{\tilde{T}_{k,l}(n)}}, \tag{25}$$

$$\eta_{k,l,n}^{\texttt{M}} = 11\sqrt{\frac{2\mathbb{V}_{\tilde{\mathcal{I}}_{k,l}(n)}^{\texttt{M}}[V_k(t_l)] \log(n^\alpha)}{\tilde{T}_{k,l}(n)}}. \tag{26}$$

In this way, the inequality $\widehat{r}_{k,l,n}^{\texttt{M}} + c_{k,l,n}^{\texttt{M}} > r_k(t_l)$ holds with high probability for sufficiently large $\tilde{T}_{k,l}(n)$. Therefore, we can construct a UCB-RM algorithm denoted as $\pi^{\texttt{M}}$ (Altorighm 2). At each decision step, UCB-RM Algorithm makes a selection for the $(n+1)$-th arm pull using the following criterion:

$$(I_{n+1}, \nu_{n+1}) \in \underset{(k,l) \in [K] \times [L]}{\arg\max} \big\{\widehat{r}_{k,l,n}^{\texttt{M}} + c_{k,l,n}^{\texttt{M}}\big\}. \tag{27}$$

In the following theorem, we analyze the performance of the UCB-RM Algorithm.

**Theorem 1** (Regret Upper Bound for UCB-RM). *For any arm $k$ and restart time $t_l$, let $\Delta_{k,l} = r_* - r_k(t_l)$ and*

$$\mathbb{C}_{k,l} = \mathbb{C}^2\big(U_{k,1}(t_l)\big) + \mathbb{C}^2\big(V_{k,1}(t_l)\big), \tag{28}$$

$$\mathbb{K}_{k,l} = \mathbb{K}\big(U_{k,1}(t_l)\big) + \mathbb{K}\big(V_{k,1}(t_l)\big), \tag{29}$$

$$z(\beta) = \max\Big\{2\sqrt{2}\frac{(1+\beta)^2}{(1-\beta)^3}, \frac{1}{\beta}\Big\}, \ \beta \in (0, 1), \tag{30}$$

where $\mathbb{C}(X)$ and $\mathbb{K}(X)$ is the coefficient of variation and kurtosis of a random variable $X$, respectively (Gallager, 2013). Then the regret under the UCB-RM Algorithm is bounded as follows:

$$\texttt{REG}_{\pi^{\texttt{M}}}(\tau) \leq \sum_k 3\alpha \min\{\xi_k^{\texttt{M}}, \tilde{\xi}_k^{\texttt{M}}\} \log(\tau) + O(KL),$$

where for each $k \in [K]$ and some constant $\zeta > 0$, $\alpha > 2$,

$$\xi_k^{\texttt{M}} = \sum_{l=1}^{L} \left\{ 11^2 z^2(\beta) \mathbb{C}_{k,l} \frac{r_*^2}{\Delta_{k,l}} + 1024 (\mathbb{K}_{k,l} + \zeta) \Delta_{k,l} \right\}, \tag{31}$$

$$\tilde{\xi}_k^{\texttt{M}} = \max_l \Delta_{k,l} \cdot \max_l \left\{ 11^2 z^2(\beta) \mathbb{C}_{k,l} \frac{r_*^2}{\Delta_{k,l}^2} + 1024 (\mathbb{K}_{k,l} + \zeta) \right\}. \tag{32}$$

Note that the regret upper bound in Theorem 1 is dependent only on the first- and second-order moments of $X_{k,n}$ and $R_{k,n}$. There is no dependency on $t_L$. Therefore, the UCB-RM Algorithm is efficient for the cases with very large (potentially infinite) $t_L$. Also, compared with the median-based algorithm UCB-BwI in (Cayci et al., 2019), UCB-RM Algorithm does not require any prior knowledge of the statistics like moments. Instead, it uses empirical estimates for reward rate, mean completion time, and variances.

### 5.1.2   UCB-RB Algorithm

In many cases, the optimal restart time is finite for all arms, i.e., $t_k^* < \infty, \forall k$. Therefore, the action set $\mathbb{T}$ can be localized around the potential optimal restart times and $t_L$ can have a finite value. Under this scenario, the support set of the completion times $U_{k,n}(t)$ is small for all $t \in \mathbb{T}$, which enables the use of fast estimation techniques. Based on this, we propose an algorithm with better performance based on empirical Bernstein inequality inspired by the UCB-B2 Algorithm in the classical stochastic bandits with knapsacks setting (Cayci et al., 2020). We denote the new algorithm as UCB-RB ($\pi^{\texttt{B}}$). Since it uses empirical estimation, it inherently assumes that $t_L < \infty$.

For any $(k, t_l)$ pair, we define the empirical reward rate as:

$$\widehat{r}_{k,l,n} = \frac{\widehat{\mathbb{E}}_{\tilde{\mathcal{I}}_{k,l}(n)}[V_k(t_l)]}{\widehat{\mathbb{E}}_{\tilde{\mathcal{I}}_{k,l}(n)}[U_k(t_l)]}, \tag{33}$$

For $\alpha > 2$ and $\beta \in (0, 1)$, we can construct the confidence radii as

$$c_{k,l,n} = \frac{(\beta+1)^2}{1-\beta} \frac{\eta_{k,l,n} + \widehat{r}_{k,l,n} \epsilon_{k,l,n}}{\widehat{\mathbb{E}}_{\tilde{\mathcal{I}}_{k,l}(n)}[U_k(t_l)]}, \tag{34}$$

where the parameters are defined as

$$\epsilon_{k,l,n} = \frac{3 t_l \log(n^\alpha)}{\tilde{T}_{k,l}(n)} + \sqrt{\frac{2 \mathbb{V}_{\tilde{\mathcal{I}}_{k,l}(n)}(U_k(t_l)) \log(n^\alpha)}{\tilde{T}_{k,l}(n)}},$$

$$\eta_{k,l,n} = \frac{3 \log(n^\alpha)}{\tilde{T}_{k,l}(n)} + \sqrt{\frac{2 \mathbb{V}_{\tilde{\mathcal{I}}_{k,l}(n)}(V_k(t_l)) \log(n^\alpha)}{\tilde{T}_{k,l}(n)}}.$$

Then, the controller under UCB-RB makes a decision at $(n+1)$-th using the following criterion:

$$(I_{n+1}, \nu_{n+1}) \in \underset{(k,t_l) \in [K] \times \mathbb{T}}{\arg\max} \left\{ \widehat{r}_{k,l,n} + c_{k,l,n} \right\}.$$

The following theorem provides problem-dependent regret upper bounds for the UCB-RB Algorithm.

---

**Algorithm 2:** Online Learning Algorithms for Finite Set of Restart Times
UCB-RM ($\pi^{\mathtt{M}}$) and UCB-RB ($\pi^{\mathtt{B}}$)

---

$n = 1, i = 1, S_0 = 0$;
// Begin initialization for `init` trials
**while** $i \leq$ `init` & $S_{n-1} \leq B$ **do**
    **for** $k \in [K], l \in [L]$ **do**
        $(I_n, \nu_n) = (k, t_l)$;
        $S_n = U_{k,n}(t_l)$;
        Obtain reward $V_{k,n}(t_l)$;
        $n = n + 1$;
    $i = i + 1$;
**while** $S_{n-1} \leq B$ **do**
    // If using UCB-RM Algorithm
    $(I_n, \nu_n) = \arg\max\limits_{(k, t_l)} \{\widehat{r}^{\mathtt{M}}_{k,l,n-1} + c^{\mathtt{M}}_{k,l,n-1}\}$
    // If using UCB-RB Algorithm
    $(I_n, \nu_n) = \arg\max\limits_{(k, t_l)} \{\widehat{r}_{k,l,n-1} + c_{k,l,n-1}\}$
    **if** $X_{I_n,n} \leq \nu_n$ **then**
        $S_n = S_{n-1} + X_{I_n,n}$;
        Obtain reward $R_{I_n,n}$;
    **else**
        $S_n = S_{n-1} + \nu_n + C_{k_n}(\nu_n)$;
    Update the estimates for $l : t_l \leq \nu_n$;
    Add $n \mapsto \tilde{\mathcal{I}}_{k,l}(n)$ for all $l : t_l \leq \nu_n$;
    $n = n + 1$;

---

**Theorem 2** (Regret Upper Bound for UCB-RB). *For any arm $k$ and restart time $t_l$, let $\Delta_{k,l} = r_* - r_k(t_l)$, $\mu_{k,l} = \mathbb{E}[U_{k,1}(t_l)]$ and*

$$\mathbb{B}_{k,l} = \frac{1}{\mathbb{E}[V_{k,1}(t_l)]} + \frac{2t_l}{\mathbb{E}[U_{k,1}(t_l)]}, \tag{35}$$

*Then, the regret under the UCB-RB Algorithm is bounded as follows:*

$$\mathtt{REG}_{\pi^{\mathtt{B}}}(\tau) \leq \sum_k 3\alpha \min\{\xi_k, \xi_k'\} \log(\tau) + O(KL),$$

*where for each $k \in [K]$ and some $\alpha > 2$,*

$$\xi_k = \sum_{l=1}^{L} \Delta_{k,l} \cdot \mu_{k,l} \cdot \left\{ z^2(\beta)\mathbb{C}_{k,l} \frac{r_*^2}{\Delta_{k,l}^2} + \frac{2z(\beta)r_*\mathbb{B}_{k,l}}{\Delta_{k,l}} + 8\big(\mathbb{K}_{k,l} + \mathbb{B}_{k,l}^2\big) \right\}, \tag{36}$$

$$\tilde{\xi}_k = \max_l\{\Delta_{k,l} \cdot \mu_{k,l}\} \cdot \max_l \left\{ z^2(\beta)\mathbb{C}_{k,l} \frac{r_*^2}{\Delta_{k,l}^2} + 2z(\beta)\mathbb{B}_{k,l} \frac{r_*}{\Delta_{k,l}} + 8\big(\mathbb{K}_{k,l} + \mathbb{B}_{k,l}^2\big) \right\}, \tag{37}$$

*with $\mathbb{C}_{k,l}$, $\mathbb{K}_{k,l}$, and $z(\beta)$ defined in Theorem 1.*

*(Comparison between UCB-RM and UCB-RB)* Comparing the coefficients of the $\log(\tau)$ term in Theorem 2 and Theorem 1, we observe that the UCB-RM Algorithm suffers from a considerably large scaling coefficient. This suggests that if the optimal restart times are known to be small, then the UCB-RB Algorithm is more efficient than the UCB-RM Algorithm.

On the other hand, the regret upper bound in Theorem 2 grows at a rate $O(t_L^2 \log(\tau))$, where the constant additive term is independent of $\tau$ and $t_L$ if $\mathbb{E}[X_{k,1}^p] < \infty$ for some $p > 2$. Therefore, if the optimal restart

time can take on a large value, then the regret performance deteriorates significantly. This dependence on $t_L$ stems from the nature of the empirical mean estimator used for estimating the reward rate, and it is inevitable (Audibert et al., 2009). Therefore, the UCB-RB Algorithm is suitable only for the cases where the restart times are small.

*(Importance of Information Structure)* For any arm $k$, $\xi_k$ corresponds to the coefficient without using the information structure, grows linearly with $L$, and shows that the dependence of the regret on $\Delta_{k,l}$ is $O(1/\Delta_{k,l})$. On the other hand, $\tilde{\xi}_k$ reflects the effect of exploiting the structure, and it is usually much lower than $\xi_k$. Hence, ignoring the constants, let $\Delta_k = \min_l \Delta_{k,l}, \forall l, k \neq k^*$, we have the following order result for the regret:

$$\texttt{REG}_{\texttt{RB}}(B) = O\Big(t_L^2 \big(\sum_{k \neq k^*} \frac{1}{\Delta_k} + \frac{1}{\min_l \Delta_{k^*,l}} + \sum_k \max_l \Delta_{k,l}\big) \log(\tau)\Big),$$

since $\mathbb{K}(Z) \leq b^2$ for a bounded random variable $Z \in [0,b]$. In other words, as a result of exploiting the information structure, the effect of large $L$ on the regret is eliminated.

*Proof Sketch:* The proof of Theorem 1 and Theorem 2 follows a similar strategy as (Cayci et al., 2019), and can be found in detail in the Appendix. We will provide a proof sketch here. The main challenge in the proof is two-fold: (1) analyzing the effect of using empirical estimates, and (2) finding tight upper and lower bounds for the expectation of the total reward $\texttt{REW}_\pi(\tau)$, which is a controlled and stopped random walk with non-i.i.d. increments.

For any $(k, t_l) \in [K] \times \mathbb{T}$, let $T_{k,l}(n)$ be the number of times the controller makes the decision $(k, t_l)$ in the first $n$ stages, and $\Delta_{k,l}$ be the expected 'regret per unit time' if $(k, t_l)$ is chosen. Then, by using tools from renewal theory and martingale concentration inequalities, we express the regret as follows:

$$\texttt{REG}_{\pi^{\texttt{B}}}(\tau) \leq \sum_{(k,l):\Delta_{k,l}>0} O(1) \, \mathbb{E}[T_{k,l}(n_0(\tau))]\mathbb{E}[U_{k,1}(t_l)]\Delta_{k,l} + O(KL),$$

where $n_0(\tau)$ is a high-probability upper bound for $N_{\pi^{\texttt{B}}}(\tau)$, the total number of pulls in $[0,\tau]$. By using a clean-event bandit analysis akin to (Audibert et al., 2009) to bound $\mathbb{E}[T_{k,l}(n_0(\tau))]$, we prove the theorem.

Note that the UCB-RM and UCB-RB Algorithm makes use of the median-of-mean method and empirical estimates to achieve regret performance guarantee without any prior knowledge, and we devise novel tools to analyze the impact of using these estimation methods on the regret. In the next section, we will develop a learning algorithm for the case $\mathbb{T}$ is continuous.

## 5.2 Continuous Set of Restart Times: UCB-RC

When the set of restarting times $\mathbb{T}$ is continuous, one common technique is to discretize the set. The key question is how to design a good discretization method with provably good regret performance. In this section, we propose an algorithm called UCB-RC for smooth and unimodal $r_k(t)$ that has $\sqrt{\tau \log \tau}$ regret guarantee.

For the notation simplicity, we will consider learning the optimal restart time for a single arm. The extension to $K > 1$ is straightforward. Specifically, we make the following assumptions on the action set $\mathbb{T}$ and reward rate function $r_1(t)$.

**Assumption 2.** *The set of restart times $\mathbb{T}$ and reward rate $r_1(t)$ satisfy the following properties:*

(i) *Compactness: $\mathbb{T}$ is a compact subset of $\mathbb{R}_+$:*

$$\mathbb{T} = [t_{min}, t_{max}], \tag{38}$$

*where $0 < t_{min} \leq t_{max} < \infty$. Furthermore, $\mathbb{T}$ satisfies Assumption 1 for some $\epsilon, \mu_* > 0$ for efficient estimation of $r_1(t)$.*

(ii) *Unimodality: There is an optimal restart time $t_1^* \in [t_{min}, t_{max}]$ such that*

$$r_* = r_1(t_1^*) \geq r_1(t),$$

*for all $t \in \mathbb{T}\backslash\{t_1^*\}$.*

*(iii) Smoothness: There exists $\delta_0 > 0, a_1 > 0, a_2 > 0,$ and $q > 1$ such that:*

- *For all $t, t' \in [t_1^*, t_1^* + \delta_0]$ (or $[t_1^* - \delta_0, t_1^*]$), the following holds:*

$$a_1 |t - t'|^q \leq |r_1(t) - r_1(t')|$$

- *For some $\delta \leq \delta_0$, if $|t - t_1^*| \leq \delta$, we have:*

$$r_* - r_1(t) \leq a_2 \delta^q.$$

Note that Assumption 2 is satisfied for a broad class of distributions. For example, if $X_{k,n}$ and $R_{k,n}$ are independent and $X_{k,n}$ has a uniform, exponential, or Pareto distribution, then the conditions are trivially satisfied.

Under Assumption 2, we design the UCB-RC algorithm (denoted as $\pi^c$) based on the UCB-RB Algorithm and the bandit optimization methodology in (Combes & Proutiere, 2014; Combes et al., 2015). As we will see, making use of the information structure is necessary to achieve a good regret performance.

---

**Algorithm 3:** Online Learning Algorithms for Continuous Set of Restart Times
UCB-RC ($\pi^c$)

---

//(1) Discretize the set of restarting times
Set the step size for discretization

$$\delta = \left( \sqrt{\log(\tau)/\tau} \right)^{1/q}$$

For $\texttt{rad}(\mathbb{T}) = t_{max} - t_{min}$, let $L(\delta) = \texttt{rad}(\mathbb{T})/\delta$
Obtain the discretized set $\mathbb{T}_Q = \{t_1, t_2, \ldots, t_{L(\delta)}\}$, where

$$t_l = t_{min} + (l - 1) \cdot \lceil 1/\delta \rceil, \ l = 1, 2, \ldots, L(\delta).$$

//(2) Run the algorithm for discrete set
Run the UCB-RB Algorithm over the action set $\mathbb{T}_Q$.

---

The following theorem provides a regret bound for the UCB-RC Algorithm.

**Theorem 3** (Regret Upper Bound for UCB-RC)**.** *Under Assumption 2, the regret for the UCB-RC Algorithm satisfies the following asymptotic upper bound:*

$$\limsup_{\tau \to \infty} \frac{\texttt{REG}_{\pi^c}(\tau)}{\sqrt{\tau \log(\tau)}} \leq 6^q a_2 \frac{\mathbb{E}[U_{1,1}]}{\mu_*} + \frac{3\alpha q}{a_1(q-1)} \mathbb{C}^\star z^2(\beta) r_*^2, \tag{39}$$

*where*

$$\mathbb{C}^\star = \left( \frac{\mathbb{E}[U_{1,1}^2]}{\mu_*^2} + \frac{\mathbb{E}[V_{1,1}^2]}{\epsilon^2 \left( \mathbb{E}[V_{1,1}] \right)^2} \right).$$

 *and $z(\beta)$ is defined in Theorem 1.*

*Proof Sketch:* The detailed proof of Theorem 3 can be found in the Appendix. Here we give a sketch of the proof. First, note that the UCB-RC Algorithm is based on discretizing the decision set $\mathbb{T}$, and running the UCB-RB Algorithm over the discretized decision set $\mathbb{T}_Q$. Therefore, we need to show that the step size $\delta$ chosen in UCB-RC can make sure the optimal reward rate over $\mathbb{T}_Q$ is close enough to $r_*$. Second, we also need to show that the number of quantization levels is kept sufficiently small. Finally, under the compactness and smoothness assumptions summarized in Assumption 2, the regret upper bound can be obtained.

*(Importance of Information Structure)* The UCB-RC Algorithm is based on an extension of the UCB-type algorithm for unimodal discrete-time stochastic bandits proposed in (Combes & Proutiere, 2014; Combes et al., 2015). A straightforward extension of the algorithm in (Combes et al., 2015) would yield $O(\log(\tau)\sqrt{\tau})$ regret. However, the UCB-RC Algorithm achieves $O(\sqrt{\tau \log(\tau)})$ regret in this case. The order reduction by a factor of $O(\sqrt{\log(\tau)})$ is because UCB-RC incorporates the information structure that stems from the right-censored feedback, as discussed in Section 2.1.

# 6 Numerical Experiments

In this section, we evaluate the performance of the proposed learning algorithms for three different applications: algorithm portfolios for SAT solvers, task scheduling in distributed networks, and tuning stopping time for neural network training.

For these experiments, the restart times are finite. Therefore, we used the UCB-RB Algorithm with $\alpha = 2.01$, $(1 + \beta)^2/(1 - \beta) = 1.01$. For initialization, the controller performed 40 trials for each $(k, t_l)$ decision. The average running time per decision step is around 120 ms on a personal computer. For comparison, we used Luby restart strategy with various hand-tuned base cutoff values as a benchmark (Luby et al., 1993). Note that without any prior knowledge, the performance of Luby restart strategy is hit-or-miss, depending on how close the chosen (guessed) base cutoff value is to the optimal restart time.

From the simulations, we will see that our setup is general enough that it can be applied to various different fields. It is worth noting that there may exist fine-tuned algorithms for specific applications that outperform both Luby and our algorithms. However, as a general framework, our algorithms do not require prior knowledge of the system and can optimize the cumulative reward while finding the optimal restart time.

## 6.1 Algorithm Portfolios for SAT Solvers

In this experiment, we consider a similar setup as the SAT Competition: for a given time interval $[0, \tau]$, the performance metric is the number of solved problem instances, thus there is a unit reward for each successful assignment, i.e., $R_{1,n} = 1$. We evaluated the performance of the meta-algorithms over the widely used Uniform Random-3-SAT benchmark set of satisfiable problem instances in the SATLIB library (Hoos & Stützle, 2000). In the dataset `uf-100-430`, there are 1000 uniformly generated problem instances with 100 variables and 430 clauses, therefore it is reasonable to assume i.i.d. completion times. Each successful assignment yields a reward $R_{1,n} = 1$. We select $\mathbb{T} = \{10^{-0.5+i\times0.125} : i = 0, 1, \ldots, 8\}$.

The empirical reward rate as a function of the restart time for the data set is given in Figure 2a. It is observed that the controlled restarts are essential for optimal performance, in accordance with the power-law completion time distributions (Gomes et al., 1998) and Proposition 6. This implies that our design is suitable for this scenario. The number of solved problem instances for different $\tau$ values are given in Figure 2b. We observe that the UCB-RB Algorithm learns the optimal restart strategy fast without any prior information, and its performance outperforms alternatives, especially at large time horizons. On the other hand, Luby restart strategy, which requires the base cutoff value as an input, is prone to perform badly with inaccurate prior information. Even if a genie provides a well-chosen base cutoff value to Luby restart strategy, it is outperformed linearly over $\tau$ by the UCB-RB Algorithm, which requires no prior information.

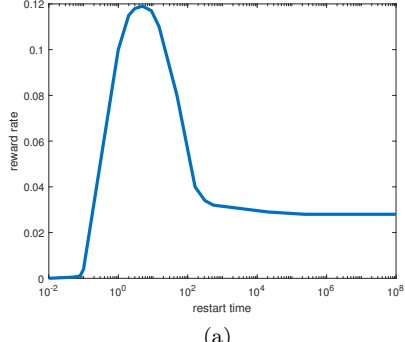

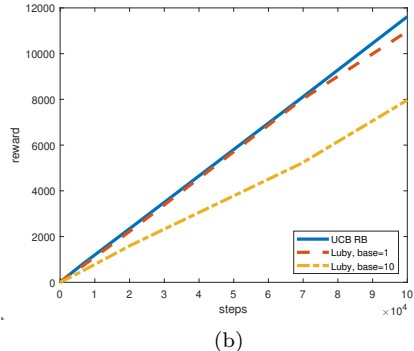

(a)    (b)

Figure 2: Performance of the restart strategies on the Random-3-SAT data set `cbs-k3`.

## 6.2 Task Scheduling in Distributed Networks

In this experiment, We considered a task scheduling problem for a single machine during its busy period. The machine selected will have a random completion time for the task during which the controller can decide whether to interrupt the machine. If the machine completes the task before interruption we receive a reward of 1 otherwise the reward is 0. The objective of the controller is to maximize the cumulative reward under a limited time $\tau$.

It has been shown that the task completion time in many distributed networks has a heavy-tailed distribution (Harchol-Balter, 1999; Xu et al., 2024), therefore our restarting strategy fits into this setup. Using the parameters defined in (Harchol-Balter, 1999), we plot the reward rate in Figure 3a. As we can see, the heavy-tailed distribution fits well with our setup. We select $\mathbb{T} = \{10^{0.5+i\times 0.5} : i = 0, 1, \dots, 8\}$. The rewards for different budgets are given in Figure 3b. Again, the UCB-RB algorithm outperforms the best Luby algorithm which requires a manual tuning on the base parameter.

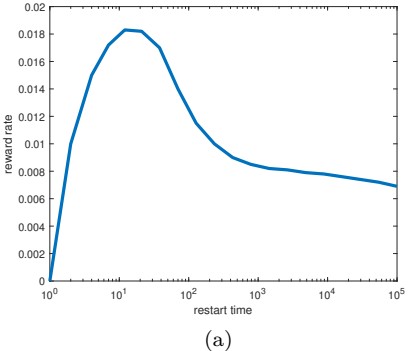
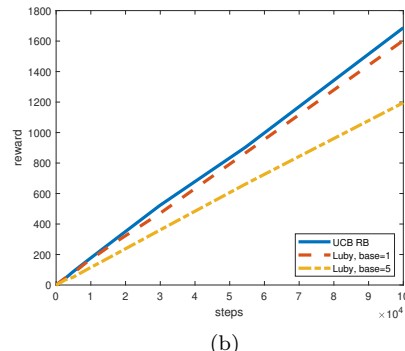

(a)  (b)

Figure 3: Performance UCB-RB algorithms for task scheduling in distributed networks.

## 6.3 Tuning the Stopping time of Neural Networks

In this experiment, we consider a setup where we need to train a set of neural networks to solve a sequence of tasks (with similar statistics). Each time, we start training a neural network with random initialization. The controller will decide when to stop the training to evaluate the test results. If the test result reaches a predetermined level we receive a reward 1 otherwise the reward is 0. Under a limited time $\tau$ (which is equivalent to training resources like GPU time), we would maximize the cumulative reward.

Specifically, in our setup, we trained a RESNET-16 over the CIFAR-10 dataset where we used $80 : 20$ split for training set and testing set. To emulate the sequence of tasks, each time we randomly sample 10 percent of the training data as the new tasks. The learning rate is set to 0.001 using an SGD optimizer with a batch size of 64. We choose 0.9 as the reward threshold. As shown in Figure 4a, the empirical reward rate suggests a restarting strategy in this work would be a good fit. We select $\mathbb{T} = \{10^{2+i\times 0.25} : i = 0, 1, \dots, 8\}$. Finally, we plot the reward against the number of allowed steps in Figure 4b. As expected, the UCB-RB algorithm can learn to optimize the cumulative reward without prior knowledge of the system statistics.

## 7 Conclusions

In this paper, we considered the continuous-time bandit learning problem with controlled restarts, and presented a principled approach with rigorous performance guarantees. For correlated and potentially heavy-tailed completion time and reward distributions, we proposed a simple, intuitive, and near-optimal offline policy with $O(1)$ optimality gap, and characterized the nature of optimal restart strategies by using this approximation. For online learning, we considered discrete and continuous action sets, and proposed bandit algorithms that exploit the statistical structure of the problem to achieve tight performance guarantees.

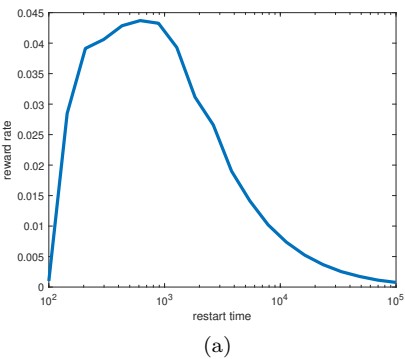 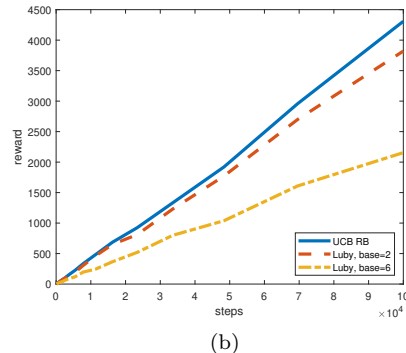

Figure 4: Performance of UCB-RB on task solving when training a neural network.

In addition to the theoretical analysis, we evaluated the numerical performance of various applications and observed that the learning solution proposed in this paper outperforms Luby restart strategy with no prior information. Another interesting direction is to derive regret lower bounds for this problem. This is complicated by the stochastic nature of the stopping time of the decision-making process, and also due to the use of empirical estimators for higher-order moments in the UCB-type algorithms. This makes the lower bound derivation significantly more complicated compared to the traditional stochastic bandit settings, and therefore we leave it as a future research problem. Finally, relaxing the assumptions, such as using random resetting times and dynamic discretization parameters, is among interesting future research directions.

## Acknowledgments

This work was funded by the Federal Ministry of Education and Research (BMBF) and the Ministry of Culture and Science of the German State of North Rhine-Westphalia (MKW) under the Excellence Strategy of the Federal Government and the Länder. Atilla Eryilmaz's research was funded by NSF grants: NSF AI Institute (AI-EDGE) 2112471, CNS-NeTS-2106679, CNS-NeTS-2007231, the ONR Grant N00014-19-1-2621; and the ARO Grant W911NF-24-1-0103.

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

## A    Appendix

## B    Proof of Proposition 6

*Proof.* It is optimal to stop if and only if the reward rate of interruption is greater than the reward rate without interruption, i.e. $r_k(t) - \mathbb{E}[R_{k,1}]/\mathbb{E}[X_{k,1}]$.

$$r_k(t) - \mathbb{E}[R_{k,1}]/\mathbb{E}[X_{k,1}] = \frac{\mathbb{E}[V_{k,1}(t)] \cdot \frac{\mathbb{E}[X_{k,1}]}{\mathbb{E}[R_{k,1}]} - \mathbb{E}[U_{k,1}(t)]}{\mathbb{E}[U_{k,1(t)}]\mathbb{E}[X_{k,1}]/\mathbb{E}[R_{k,1}]}$$

Using the identities

$$\mathbb{E}[U_{k,1(t)}] = \mathbb{E}[X_{k,1} - (t + C_k(t))|X_{k,1} > t](1 - \mathbb{P}(X_{k,1} \leq t)) - \mathbb{E}[X_{k,1}]$$
$$\mathbb{E}[V_{k,1}(t)] = \mathbb{E}[R_{k,1}] - \mathbb{E}[R_{k,1}|X_{k,1} > t](1 - \mathbb{P}(X_{k,1} \leq t))$$

we have

$$r_k(t) - \mathbb{E}[R_{k,1}]/\mathbb{E}[X_{k,1}]$$
$$= (1 - \mathbb{P}(X_{k,1} \leq t))\frac{\mathbb{E}[X_{k,1} - (t + C_k(t))|X_{k,1} > t] - \mathbb{E}[R_{k,1}|X_{k,1} > t]\frac{\mathbb{E}[X_{k,1}]}{\mathbb{E}[R_{k,1}]}}{\mathbb{E}[U_{k,1(t)}]\mathbb{E}[X_{k,1}]/\mathbb{E}[R_{k,1}]}$$

We get the results in Proposition 6 by letting the right side be greater than 0.

$\square$

## C    Concentration Inequalities for Reward Rate Estimation

In this section, we will provide tight concentration inequalities for the empirical mean estimators and median-of-mean estimators used in the main paper. First, we give a lemma that provides a basis to design these concentration inequalities.

**Lemma 1** (Proposition 2, (Cayci et al., 2020))**.** *Consider a pair of parameters $\mu_X, \mu_R > 0$, and their estimators $\widehat{\mu}_X$ and $\widehat{\mu}_R$, respectively. Let $\widehat{r} = \widehat{\mu}_R/\widehat{\mu}_X$ and $r = \mu_R/\mu_X$. Then, for any $\beta \in (0, 1)$, we have the following inequality:*

$$\mathbb{P}\left(|\widehat{r} - r| > \frac{(1 + \beta)^2}{1 - \beta}\frac{\eta + \widehat{r} \cdot \epsilon}{\widehat{\mu}_X}\right) \leq \mathbb{P}(|\mu_X - \widehat{\mu}_X| > \epsilon) + \mathbb{P}(|\mu_R - \widehat{\mu}_R| > \eta), \tag{40}$$

*for any $\eta \leq \beta\mu_R$, and $\epsilon \leq \beta\mu_X$.*

*Proof.* Let

$$A(\epsilon, \eta) = \{|\mu_X - \widehat{\mu}_X| \leq \epsilon\} \cap \{|\mu_R - \widehat{\mu}_R| \leq \eta\},$$

be the high-probability event. Then, by Proposition 1 in (Cayci et al., 2020), we have the following set relation:

$$A(\epsilon, \eta) \subset \left\{|\widehat{r} - r| \leq \frac{\eta + r\epsilon}{\mu_X - \epsilon}\right\}.$$

For any $\beta \in (0,1)$, if $\epsilon \leq \beta\mu_X$ and $\eta \leq \beta\mu_R$ is satisfied, then we have:

$$\frac{\eta + r\epsilon}{\mu_X - \epsilon} \leq \frac{(1+\beta)^2}{1-\beta} \frac{\eta + \widehat{r}\epsilon}{\widehat{\mu}_X},$$

within the set $A(\epsilon, \eta)$. By taking the compliment and using union bound, we obtain the result. $\square$

By using Lemma 1, we can prove tight concentration bounds for the renewal rate as follows.

**Proposition 7** (Concentration inequalities for reward rate). *Let $\{(X_n, R_n) : n \geq 1\}$ be a renewal reward process.*

1. *If $X_n \in [0, a]$ and $R_n \in [0, b]$, let*

$$\widehat{r}_n = \frac{\widehat{\mathbb{E}}_{[n]}[X]}{\widehat{\mathbb{E}}_{[n]}[R]},$$

   *and*

$$\epsilon_n(\delta) = \sqrt{\frac{2\mathbb{V}_{[n]}(X)\log(1/\delta)}{n}} + \frac{3a\log(1/\delta)}{n},$$

$$\eta_n(\delta) = \sqrt{\frac{2\mathbb{V}_{[n]}(R)\log(1/\delta)}{n}} + \frac{3b\log(1/\delta)}{n}.$$

   *Then, for any $\beta \in (0,1)$ and $\delta \in (0,1)$, we have:*

$$\mathbb{P}\left(|\widehat{r}_n - r| > \frac{(1+\beta)^2}{1-\beta} \cdot \frac{\eta_n(\delta) + \widehat{r}_n\epsilon_n(\delta)}{\widehat{\mathbb{E}}_{[n]}[X]}\right) \leq 12\delta,$$

   *for any $n \geq 8\left(\mathbb{K}(X_1) + \mathbb{K}(R_1) + \frac{a^2}{Var(X_1)} + \frac{b^2}{Var(R_1)}\right) + 3\left(\frac{\mathbb{C}^2(X_1)}{\beta^2} + \frac{a}{\beta\mathbb{E}[X_1]} + \frac{\mathbb{C}^2(R_1)}{\beta^2} + \frac{b}{\beta\mathbb{E}[R_1]}\right).$*

2. *Consider a renewal reward process such that $\mathbb{E}[X_1^4] < \infty$ and $\mathbb{E}[R_1^4] < \infty$. For $m = \lfloor 3.5\log(1/\delta)\rfloor + 1$, let $G_1, G_2, \ldots, G_m$ be a partition of $[n]$ such that $|G_j| = \lfloor n/m\rfloor$. Then, we define the median-based mean and variance estimators as follows:*

$$\mathbb{M}_{[n]}(X) = \texttt{median}\{\widehat{\mathbb{E}}_{G_1}[X], \widehat{\mathbb{E}}_{G_2}[X], \ldots, \widehat{\mathbb{E}}_{G_m}[X]\},$$

$$\mathbb{V}_{[n]}^{\texttt{M}}(X) = \texttt{median}\{\mathbb{V}_{G_1}[X], \mathbb{V}_{G_2}[X], \ldots, \mathbb{V}_{G_m}[X]\}.$$

   *Let*

$$\widehat{r}_n^{\texttt{M}} = \frac{\widehat{\mathbb{E}}_{[n]}^{\texttt{M}}[R]}{\widehat{\mathbb{E}}_{[n]}^{\texttt{M}}[X]},$$

   *and*

$$\epsilon_n^{\texttt{M}}(\delta) = 11\sqrt{\frac{2\mathbb{V}_{[n]}^{\texttt{M}}(X)\log(1/\delta)}{n}},$$

$$\eta_n^{\texttt{M}}(\delta) = 11\sqrt{\frac{2\mathbb{V}_{[n]}^{\texttt{M}}(R)\log(1/\delta)}{n}}.$$

*Then, for any $\beta \in (0,1)$ and $\delta \in (0,1)$, we have:*

$$\mathbb{P}\Big(|\widehat{r}_n^{\mathtt{M}} - r| > \frac{(1+\beta)^2}{1-\beta} \cdot \frac{\eta_n^{\mathtt{M}}(\delta) + \widehat{r}_n^{\mathtt{M}}\epsilon_n^{\mathtt{M}}(\delta)}{\widehat{\mathbb{E}}_{[n]}^{\mathtt{M}}[X]}\Big) \le 16.8\delta,$$

*for $n \ge 1024\big(\mathbb{K}(X_1) + \mathbb{K}(R_1) + \frac{\mathbb{C}^2(X_1) + \mathbb{C}^2(R_1)}{\beta^2} + \zeta\big)$ for some $\zeta > 0$.*

*Proof.*      1. If $n \ge 8\big(\mathbb{K}(X_1) + \frac{a^2}{Var(X_1)}\big)$, then the following holds with probability at least $1 - 4\delta$:

$$|\mathbb{V}_{[n]}(X_1) - Var(X_1)| \le Var(X_1)/2,$$

where $\mathbb{K}(X) = \frac{\mathbb{E}|X - \mathbb{E}X|^2}{Var^2(X)}$ is the kurtosis of a random variable $X$. Therefore, with probability at least $1 - 4\delta$, we have the following:

$$\sqrt{\frac{3Var(X_1)\log(1/\delta)}{n}} + \frac{3a\log(1/\delta)}{n} \le \beta\mathbb{E}[X_1].$$

Hence, if $n \ge 3\big(\frac{\mathbb{C}^2(X_1)}{\beta^2} + \frac{a}{\beta\mathbb{E}[X_1]}\big)$ also holds, then the above inequality is automatically satisfied, where $\mathbb{C}(X) = \sqrt{Var(X)}/\mathbb{E}[X]$ is the coefficient of variation. Therefore, if

$$n \ge 8\Big(\mathbb{K}(X_1) + \mathbb{K}(R_1) + \frac{a^2}{Var(X_1)} + \frac{b^2}{Var(R_1)}\Big) + 3\Big(\frac{\mathbb{C}^2(X_1)}{\beta^2} + \frac{a}{\beta\mathbb{E}[X_1]} + \frac{\mathbb{C}^2(R_1)}{\beta^2} + \frac{b}{\beta\mathbb{E}[R_1]}\Big),$$

then we have

$$\epsilon_n(\delta) \le \beta\mathbb{E}[X_1],$$
$$\eta_n(\delta) \le \beta\mathbb{E}[R_1],$$

with probability at least $1 - 8\delta$. Then we use Lemma 1 in conjunction with empirical Bernstein inequality (see (Audibert et al., 2009)) to conclude the proof.

2. The proof follows from identical steps as Part 1, and uses Proposition 4.1 and Corollary 4.2 in (Minsker et al., 2015) for the concentration results.

$\square$

# D   Proof of Theorem 1 and Theorem 2

The number of trials $N_\pi(\tau)$ under an admissible policy $\pi$ is a random stopping time, which makes the regret computations difficult. The following proposition provides a useful tool for regret computations.

**Lemma 2** (Regret Upper Bounds for Admissible Policies). *Let $T_{k,l}(n)$ be the number of steps where the decision is $(k, t_l)$ in $n$ trials, and $\mu_* = \min_{k,t}\mathbb{E}[U_{k,1}(t_l)]$. The following upper bound holds for any admissible policy $\pi \in \Pi$ and $\tau > \mu_*/2$:*

$$\mathtt{REG}_\pi(\tau) \le \sum_{k,l}\mathbb{E}\Big[T_{k,l}\Big(\frac{2\tau}{\mu_*}\Big)\Big]\Delta_{k,l}\mathbb{E}[U_{k,1}(t_l)] + \frac{\exp(-\tau\mu_*/t_1^2)}{1 - \exp\big(\mu_*^2/(2t_1^2)\big)}\sum_{k,l}\Delta_{k,l}\mathbb{E}[U_{k,1}(t_l)] + r^*\Phi,$$

*where $\Phi$ is a constant.*

The proof of Lemma 2 relies on Azuma-Hoeffding inequality for controlled random walks, which can be found in (Cayci et al., 2019; 2020). Note that $2\tau/\mu_*$ is a high-probability upper bound for the total number of pulls $N_\pi(\tau)$, and $\Delta_{k,l}\mathbb{E}[U_{k,1}(t_l)]$ is the average regret per pull for a decision $(k, t_l)$. Lemma 2 implies that the expected regret after $2\tau/\mu^*$ pulls is $O(1)$.

In the following lemma, we quantify the scaling effect of using empirical estimates.

**Lemma 3.** *Under the UCB-RB Algorithm, we have the following upper bounds:*

*(i) If $k = k^*, t_l > t_{k^*}^*$ or $k \neq k^*, \forall l$, we have:*

$$\sum_{j=l}^{L} \mathbb{E}[T_{k,j}(N)] \leq 3\left(z^2(\beta)\mathbb{C}_{k,j}\frac{r_*^2}{\Delta_{k,j}^2} + 2z(\beta)\mathbb{B}_{k,j}\frac{r_*}{\Delta_{k,j}} + 8\big(\mathbb{K}_{k,j} + \mathbb{B}_{k,j}^2\big)\right)\log(N^\alpha) + O(L),$$

*(ii) If $k = k^*$ and $t_l < t_{k^*}^*$, we have $\mathbb{E}[T_{k,l}(n)] = O(1)$ for all $n$.*

*Proof.* Consider a suboptimal decision $(k, t_l)$ where either $\{k = k^*, t_l > t_k^*\}$ or $\{k \neq k^*\}$ is true, and let

$$E_{1,n} = \left\{\widehat{r}_* + c_* \leq r_*\right\}, \tag{41}$$

$$E_{2,n} = \bigcup_{j \geq l}\left\{\widehat{r}_{k,j,n} \geq c_{k,j,n} + r_k(t_j)\right\}, \tag{42}$$

$$E_{3,n} = \bigcup_{j \geq l}\{2c_{k,j,n} \geq \Delta_{k,j}\}. \tag{43}$$

Then, it is trivial to show by using contradiction that $\{(I_{n+1}, \nu_{n+1}) = (k, t_j) : j \geq l\} \subset \cup_{i=1}^{3} E_{i,n}$. In the following, we provide a sample complexity analysis for the events above.

For notational simplicity, for $j \geq l$, let $U_n := U_{k,n}(t_j)$ and $V_n := V_{k,n}(t_j)$ for all $n$. For sample size $s$ and any $\delta \in (0, 1)$, let $\widehat{r}_s = \frac{\widehat{\mathbb{E}}_{[s]}[V]}{\widehat{\mathbb{E}}_{[s]}[U]}$, and

$$\epsilon_s = \sqrt{\frac{2\mathbb{V}_{[s]}(U)\log(1/\delta)}{s}} + \frac{3t_j\log(1/\delta)}{s}, \tag{44}$$

$$\eta_s = \sqrt{\frac{2\mathbb{V}_{[s]}(V)\log(1/\delta)}{s}} + \frac{3\log(1/\delta)}{s}. \tag{45}$$

Then, by Proposition 7, the following holds with probability at least $1 - 12\delta$:

$$\left|\widehat{r}_s - r_k(t_j)\right| \leq \frac{(1+\beta)^2}{1-\beta}\frac{\eta_s + \widehat{r}_s\epsilon_s}{\widehat{\mathbb{E}}_{[s]}[U]}, \tag{46}$$

given $s$ is sufficiently large such that

$$\begin{aligned}\epsilon_s &\leq \beta\mathbb{E}[U_1], \\ \eta_s &\leq \beta\mathbb{E}[V_1].\end{aligned} \tag{47}$$

By using empirical Bernstein inequality and union bound for the empirical variance, we have the following inequalities with probability at least $1 - 8\delta$:

$$\begin{aligned}|\mathbb{V}_{[s]}(U) - Var(U_1)| &\leq Var(U_1)/2, \\ |\mathbb{V}_{[s]}(V) - Var(V_1)| &\leq Var(V_1)/2,\end{aligned}$$

if $s \geq s_{1,j}(\delta) := 8\big(\mathbb{K}(U_1) + \mathbb{K}(V_1) + \frac{4t_j^2}{Var(U_1)} + \frac{1}{Var(V_1)}\big)\log(1/\delta)$. Therefore, the condition equation 47 implies the following with probability at least $1 - 8\delta$ for $s \geq s_{1,j}(\delta)$:

$$\begin{aligned}\tilde{\epsilon}_s &:= \sqrt{\frac{3Var(U_1)\log(1/\delta)}{s}} + \frac{3t_j\log(1/\delta)}{s} \leq \beta\mathbb{E}[U_1], \\ \tilde{\eta}_s &:= \sqrt{\frac{3Var(V_1)\log(1/\delta)}{s}} + \frac{3\log(1/\delta)}{s} \leq \beta\mathbb{E}[V_1].\end{aligned} \tag{48}$$

The inequalities in equation 48 simultaneously hold if

$$s \geq s_{2,j}(\delta) := 3\Big(\frac{1}{\beta^2}\big(\mathbb{C}^2(U_1) + \mathbb{C}^2(V_1)\big) + \frac{1}{\beta}\big(\frac{t_j}{\mathbb{E}[U_1]} + \frac{1}{\mathbb{E}[V_1]}\big)\Big)\log(1/\delta).$$

Also, note that

$$\frac{(1+\beta)^2}{1-\beta}\frac{\eta_s + \widehat{r}_s\epsilon_s}{\widehat{\mathbb{E}}_{[s]}[U]} \leq \frac{(1+\beta)^2}{(1-\beta)^3}\frac{\tilde{\eta}_s + r_k(t_j)\tilde{\epsilon}_s}{\mathbb{E}[U]_1},$$

holds if equation 48 is true. Thus, by using this result, we show that

$$2\frac{(1+\beta)^2}{1-\beta}\frac{\eta_s + \widehat{r}_s\epsilon_s}{\widehat{\mathbb{E}}_{[s]}[U]} \leq 2\frac{(1+\beta)^2}{(1-\beta)^3}\frac{\tilde{\eta}_s + r_k(t_j)\tilde{\epsilon}_s}{\mathbb{E}[U]_1} \leq \Delta_{k,j},$$

with probability at least $1 - 8\delta$ if $s \geq \max\{s_{1,j}(\delta), s_{2,j}(\delta), s_{3,j}(\delta)\}$ for

$$s_{3,j}(\delta) = 3\Big(8\frac{(1+\beta)^4}{(1-\beta)^6}\frac{r_*^2}{\Delta_{k,j}^2}\big(\mathbb{C}^2(U_1) + \mathbb{C}^2(V_1)\big) + 2\frac{(1+\beta)^2}{(1-\beta)^3}\big(\frac{1}{\mathbb{E}[V_1]} + \frac{2t_j}{\mathbb{E}[U_1]}\big)\Big)\log(1/\delta).$$

In summary, the following events simultaneously hold with probability at least $1 - 12\delta$:

$$\Big|\widehat{r}_s - r_k(t_j)\Big| \leq \frac{(1+\beta)^2}{1-\beta}\frac{\eta_s + \widehat{r}_s\epsilon_s}{\widehat{\mathbb{E}}_{[s]}[U]} \leq \Delta_{k,j}/2, \tag{49}$$

if $s \geq \max\{s_{1,j}(\delta), s_{2,j}(\delta), s_{3,j}(\delta)\}$.

Note that the UCB-RB Algorithm is designed such that $s = T_{k,j}^*(n)$ and $\delta = n^{-\alpha}$. Now we will use the above analysis to provide an upper bound for $\sum_{j \geq l}\mathbb{E}[T_{k,j}(n)]$. First, let

$$A_j = \{T_{k,j}^*(n) \leq u_j\}, \ j \geq l$$

for $u_j = \max_i\{s_{i,j}(N^{-\alpha})\}$. Then, equation 49, we have the following inequality:

$$\mathbb{P}\Big(\big(\cup_{j \geq l} A_j\big) \cap \big(E_{1,n} \cup E_{2,n} \cup E_{3,n}\big)\Big) \leq 16(L-l)/n^{\alpha-1}, \tag{50}$$

where we used union bound to deal with the random sample size $T_{k,j} * (n) \leq n$ in computing probabilities. Now, recall that $T_{k,j}^*(n) = \sum_{j' \geq j} T_{k,j'}(n)$ by definition, and we have the following relation:

$$\bigcup_{j \geq l} A_j \subset \Big\{\sum_{j \geq l} T_{k,j}(n) \leq \max_{j \geq l} u_j\Big\}, \tag{51}$$

which follows from the fact that $\cup_{j \geq l} A_j = A_L \cup (A_{L-1} \cap A_L^c) \cup \ldots \cup (A_l \cap (\cup_{l < j \leq L} A_j)^c)$, and

$$A_j \cap (\cup_{j' > j} A_{j'})^c \subset A_j \cap A_{j+1}^c \subset \{T_{k,j} \leq \max\{0, u_j - u_{j+1}\}\}.$$

Therefore, we have the following inequality:

$$\sum_{j \geq l} T_{k,l}(N) \leq \max_{j \geq l} u_j + \sum_{i=\max_{j \geq l} u_j + 1}^{N} \mathbb{I}\{E_{1,i} \cup E_{2,i} \cup E_{3,i}\}.$$

Taking the expectation in the above equality, and using equation 50 and equation 51, we have the following result:

$$\mathbb{E}[\sum_{j \geq l} T_{k,l}(N)] \leq \max_{j \geq l} u_j + \frac{16(L-l)\alpha}{\alpha-2},$$

which yields the result in part (i).

For part (ii), let $(k, t_l)$ be such that $k = k^*$ and $t_l < t_k^*$. Following a similar analysis as part (i) yields $\mathbb{E}[T_{k,l}(N)] \leq O(\log(N^\alpha))$, which implies that $\mathbb{E}[T_{k^*, l_{k^*}^*}(N)] = \Omega(N - \log(N))$. Therefore, since the number of samples satisfies $\tilde{T}_{k,l}(N) \geq T_{k^*, l_{k^*}^*}(N) = \Omega(N - \log(N))$, the decision $(k^*, t_l)$ is chosen at most $O(1)$ times (Lattimore & Munos, 2014; Cayci & Eryilmaz, 2017).

Combining Lemma 3 and Lemma 2, the results of Theorem 2 follows by summarizing over all arms. The Proof of Theorem 1 are similar except for using the second part of results in Proposition 7.

$\square$

# E   Proof of Theorem 3

The proof incorporates a variant of the regret analysis for quantized continuous decision sets given in (Combes & Proutiere, 2014) into the regret analysis for budget-constrained bandits presented in Appendix D.

**Step 1.** First, we bound the regret that stems from using a quantized decision set. Recall from Proposition 4 that

$$\mathtt{OPT}(\tau) \leq \tau \cdot r_* + O(1),$$

and

$$\mathtt{OPT}_Q(\tau) \geq \tau \max_{t \in \mathbb{T}_Q} r_1(t),$$

where $\mathtt{OPT}_Q(\tau)$ is the optimal reward in the quantized decision set $\mathbb{T}_Q$, and $r_* = \max_{t \in \mathbb{T}} r_1(t)$. Then, the regret under the UCB-RC Algorithm is bounded as follows:

$$\begin{aligned}
\mathtt{REG}_{\pi^c}(\tau) &= \mathtt{OPT}(\tau) - \mathbb{E}[\mathtt{REW}_{\pi^c}(\tau)], \\
&= \mathtt{OPT}(\tau) - \mathtt{OPT}_Q(\tau) + \mathtt{OPT}_Q(\tau) - \mathbb{E}[\mathtt{REW}_{\pi^c}(\tau)], \\
&\leq \tau\big(r_* - \max_{t \in \mathbb{T}_Q} r_1(t)\big) + \mathtt{REG}_{\pi^c, Q}(\tau) + O(1),
\end{aligned}$$

where

$$\mathtt{REG}_{\pi^c, Q}(\tau) = \mathtt{OPT}_Q(\tau) - \mathbb{E}[\mathtt{REW}_{\pi^c}(\tau)],$$

is the regret under the UCB-RC Algorithm with respect to the optimal policy in the quantized decision set. By Assumption 2, we have:

$$r_* - \max_{t \in \mathbb{T}_Q} r_1(t) \leq a_2 \delta^q,$$

since $|t_1^* - \arg\max_{t \in \mathbb{T}_Q} r_1(t)| \leq \delta$. Thus, we have the following inequality:

$$\mathtt{REG}_{\pi^c}(\tau) \leq a_2 \tau \delta^q + \mathtt{REG}_{\pi^c, Q}(\tau) + O(1). \tag{52}$$

**Step 2.** After we quantify the regret from using a quantized decision set, now we bound $\mathtt{REG}_{C,Q}(\tau)$, the regret of the UCB-RC Algorithm with respect to the optimal algorithm in the quantized decision set. We first present a variation of the decomposition in equation 51.

**Claim 1.** *Let $l^* = \arg\max_j r_1(t_j)$, and $r_Q^* = \max_j r_1(t_j)$ be the optimal reward rate in the quantized decision set $\mathbb{T}_Q$. For any $l \neq l^*$, let $\Delta_{1,l} = r_Q^* - r_1(t_l)$, and*

$$z_l = 3\Big(z^2(\beta)\mathbb{C}_{1,l}\frac{(r_Q^*)^2}{\Delta_{1,l}} + O(1)\Big)\log(N^\alpha). \tag{53}$$

*Then, we have the following for any $l \leq k \leq L(\delta)$:*

$$\mathbb{E}[\sum_{j=l, j \neq l^*}^{k} \Delta_{1,j} T_{1,j}(N)] \leq z_l + \sum_{\substack{j=l+1 \\ j \neq l^*}}^{k} z_j\Big(1 - \frac{\Delta_{1,j-1}}{\Delta_{1,j}}\Big)^+ + \frac{16L(\delta)\alpha}{\alpha - 2}. \tag{54}$$

*Proof.* We have the following relation for any $l \leq k \leq L(\delta)$:

$$\cup_{j=l}^{k} A_j \subset \Big\{ \sum_{\substack{j=l, j \neq l^*}}^{k} \Delta_{1,j} T_{1,j}(N) \leq z_l + \sum_{\substack{j=l+1 \\ j \neq l^*}}^{k} z_j \Big( 1 - \frac{\Delta_{1,j-1}}{\Delta_{1,j}} \Big)^+ \Big\}, \tag{55}$$

which can be proved by induction. Note that the $O(1)$ term in the RHS of equation 53 is bounded as follows:

$$2z(\beta)\mathbb{B}_{1,l} r_Q^* + 8\big(\mathbb{K}_{1,l} + \mathbb{B}_{1,l}^2\big)\Delta_{1,l} \leq 2z(\beta)b_0 r_* + 8r_*(\kappa + b_0^2),$$

where $b_0 \geq \max_l B_{1,l}$ and $\kappa \geq \max_l \mathbb{K}_{1,l}$ are constants independent of $\delta$, but depend on $\texttt{rad}(\mathbb{T})$, $\epsilon$ and $\mu_*$ under Assumption 2.

By following the same steps as Lemma 3, the proof follows. $\square$

By equation 52 and Proposition 2, the regret under UCB-RC is bounded as follows:

$$\texttt{REG}_{\texttt{C}}(\tau) \leq a_2 \delta^q \tau + \mu \sum_{l \in [L(\delta)]} \mathbb{E}[\Delta_{1,l} T_{1,l}(2\tau/\mu_*)] + O\big(L(\delta)\big). \tag{56}$$

where $\mu = \max_{t \in \mathbb{T}} \mathbb{E}[U_{1,1}(t)]$ and $\mu_* = \min_t \mathbb{E}[U_{1,1}(t)]$.

Let the sets $A, B, D$ be defined as follows:

$$A = \{l^* - 1, l^*, l^* + 1\},$$
$$B = \{l : t_{min} + (l-1)\delta \in \mathcal{B}(t_1^*, \delta_0)\} \cap A^c,$$
$$D = [L(\delta)] \cap (A \cup B)^c,$$

where $\mathcal{B}(x, \epsilon_0)$ denotes the ball in $\mathbb{R}$ centered at $x$ with radius $\epsilon_0 > 0$.

- For any $l \in A \backslash \{l^*\}$, we have the following by Assumption 2:

$$\Delta_{1,l} \leq r_* - r_1(t_l) \leq a_2(2\delta)^q,$$

since $r_* \geq r_Q^*$ and $\big| t_1^* - \big( t_{min} + (l-1)\delta \big) \big| \leq 2\delta$. Thus, we have:

$$\sum_{l \in A \backslash \{l^*\}} \mathbb{E}\Big[\Delta_{1,l} T_{1,l}\Big(\frac{2\tau}{\mu_*}\Big)\Big] \leq a_2 \mu \cdot \frac{2\tau}{\mu_*}(2\delta)^q. \tag{57}$$

- For $l \in B$, note that

$$\Big| \big(t_{min} + (l^* - 1)\delta\big) - \big(t_{min} + (l-1)\delta\big) \Big| \geq \delta\big(|l^* - l| - 1\big),$$

which implies the following by Assumption 2:

$$\Delta_{1,l} \geq a_1 \big| |l^* - l| - 1 \big|^q \delta^q.$$

By using this result and Claim 1, we have the following bound:

$$\mathbb{E}\Big[\sum_{l \in B} \Delta_{1,l} T_{1,l}\Big(\frac{2\tau}{\mu_*}\Big)\Big] \leq \alpha \sum_{l=1}^{L(\delta)} \frac{3z^2(\beta)\mathbb{C}_{1,l} r_*^2 \log(\tau)}{a_1(l \cdot \delta)^q} + O(\log(\tau)|B|),$$

$$\leq 3\alpha z^2(\beta) r_*^2 \frac{\log(\tau)}{a_1 \delta^q} \mathbb{C}^\star \sum_{l=1}^{\infty} \frac{1}{l^q} + O(\log(\tau)|B|),$$

where $\mathbb{C}_{1,l} \leq \mathbb{C}^\star$ for all $l \in L(\delta)$ with

$$\mathbb{C}^\star = \Big( \frac{\mathbb{E}[U_{1,1}^2]}{\mu_*^2} + \frac{\mathbb{E}[V_{1,1}^2]}{\epsilon^2 \big(\mathbb{E}[V_{1,1}]\big)^2} \Big).$$

Therefore, we have the following bound:

$$\mathbb{E}\Big[\sum_{l \in B} \Delta_{1,l} T_{1,l}\Big(\frac{2\tau}{\mu_*}\Big)\Big] \leq 3\alpha z^2(\beta) r_*^2 \frac{\log(\tau)}{a_1 \delta^q} \frac{\mathbb{C}^\star q}{q-1} + O(\log(\tau)|B|). \tag{58}$$

- For $l \in D$, we have $\Big|\big(t_{min} + (l^* - 1)\delta\big) - \big(t_{min} + (l-1)\delta\big)\Big| \geq \delta_0/2$, hence the following holds by Assumption 2:

$$\Delta_{1,l} \geq a_1(\delta_0/2)^q.$$

Consequently, we have the following upper bound:

$$\mathbb{E}\Big[\sum_{l \in D} \Delta_{1,l} T_{1,l}\Big(\frac{2\tau}{\mu_*}\Big)\Big] \leq 3\alpha z^2(\beta) r_*^2 \mathbb{C}^\star \frac{\log(\tau)L(\delta)}{a_1(\delta_0/2)^q} + O(\log(\tau)|D|). \tag{59}$$

Substituting the results in equation 57, equation 58 and equation 59 into equation 56, we obtain the following upper bound:

$$\texttt{REG}_{\pi^c}(\tau) \leq a_2 \frac{2\mu}{\mu_*} \tau(3\delta)^q + \frac{3\alpha q}{q-1} \mathbb{C}^\star z^2(\beta) r_*^2 \frac{\log(\tau)}{a_1 \delta^q} + O(\log(\tau)L(\delta)). \tag{60}$$

With the choice $\delta^q = \sqrt{\log(\tau)/\tau}$, we have $\log(\tau)L(\delta) = o(\sqrt{\tau \log(\tau)})$. Therefore,

$$\limsup_{\tau \to} \frac{\texttt{REG}_{\pi^c}(\tau)}{\sqrt{\tau \log(\tau)}} \leq 6^q a_2 \frac{\mu}{\mu_*} + \frac{3\alpha q}{a_1(q-1)} \mathbb{C}^\star z^2(\beta) r_*^2.$$

