# OpenReview forum: "Budgeted-Bandits with Controlled Restarts with Applications in Learning and Computing"
_TMLR — Accepted by TMLR_

### Review · Reviewer_wmWz · 2025-02-26

**Summary Of Contributions:**

This work introduces a continuous-time bandit framework with controlled restarts, addressing the challenge of balancing task completion time and reward optimization under uncertainty. It proposes near-optimal offline policies with provable guarantees and efficient online learning algorithms that leverage problem structure, outperforming the Luby restart strategy without requiring prior information. Theoretical results and experimental results are provided.

**Audience:**

Yes

**Claims And Evidence:**

No

**Requested Changes:**

Besides the weaknesses in the above section for more clear presentation, the authors should cite more bandit work that is related with your problem setting and techniques.

1. This paper should cite some work on delayed feedback bandits, which is a hot topic and similar with the setting in this work. For delayed feedback bandit, the reward will not be immediately revealed to the agent when it is pulled, but it will be observed after some stochastic delay. Seminal works on this problem are

a. Linear bandits with stochastic delayed feedback, Vernade et al. ICML 2020;

b. Online learning with delayed feedback, Joulani et al. ICML 2013;

c. Delay and cooperation in nonstochastic bandit, Cesa-Bianchi et al. JMLR.

2. This paper should more work using restarting technique to handle the non-stochastic environment for the stochastic bandit problem. The restarting method in bandit community mostly refers to a specific technique that can handle non-stochastic bandit problems where the rewards are not sampled from a fixed distribution all the time. Seminal works using this restarting trick are:

a. A simple approach for non-stationary linear bandits, Zhao et al. AISTATS 2020;

b. Online continuous hyperparameter optimization for generalized linear contextual bandits, Kang et al. TMLR;

c. Learning to optimize under non-stationarity, Cheung et al. AISTATS 2019.

Please have a discussion on the related works mentioned in above 1, 2 in the revision (related works).

**Strengths And Weaknesses:**

Strengths
1. The work studies an interesting problem.
2. Both theoretical results and experimental results are provided.

Weaknesses:
1. The major weakness of this work is the readability. It is very hard to read this work and understand the main results of this work.

1.1 What do you mean by "continuous-time" bandit? From you regret definition in Eqn. (5), (6), (7), it seems that you are still consider discrete time for bandit pulling. Then what is the difference if the delay is 5 vs 5.5?

1.2 I do not understand Eqn. (5). Why is that the number of pulls under policy $\pi$? I understand that $U_{I_i,i}(\nu_i)$ is the total time for the $i$th pull? Is $i$th pull happening at time $i$?

1.3 The motivating examples should be moved to the introduction. I am very confused about the intuition and use case of your method at the beginning, which definitely dampers my reading.

1.4 Why do you need the bounded forth moment? In bandit literature, a random variable is heavy-tailed if it has finite $1+\epsilon$ moment where $\epsilon \in (0,1]$.

1.5 What is the assumption on the resetting time function? Why is there a resetting time intuitively?

1.6 More explanation on the reward rate will be appreciated.

2. Lower bounds should at least be discussed.
3. Some related literature should be discussed in the related work section.
4. Running time is recommended to be reported in the paper to showcase the efficiency of your method.

---

> ### Author Response · Authors · 2025-05-20
> **Response to Reviewer wmWz**
>
> We would like to thank the reviewer for the very valuable and constructive feedback, which have helped us improve the overall quality of the paper. Below, we provide our responses to all questions and comments by the reviewer. We updated the original paper, with changes highlighted in blue.
>
> 1. Thanks for the valuable suggestions on clarifying the notations and readability. These have greatly improved the quality of the paper.
>
> 1.1 In terms of the continuous bandits, we referred that the random completion time and cutoff times are continuous, as described in Equation (3). We have now made this clearer in the description. If the completion time is $5.5$ instead of $5$, the cost $U(\cdot)$ in Equation (3) would increase.
>
> 1.2 In Equation (5), the number of pulls under a policy $\pi$ is random because each arm pull takes a random completion time, and the decision-making process continues a fixed time budget $\tau$ is exceeded. More concretely, the $i$-th pull take a *random* completion time $U_{I_i, i}(\nu_i)$ as defined in Equation (3). The decision-making process continues until the total time $U_{I_1,1}(\nu_1)+U_{I_2,2}(\nu_2)+\ldots$ exceeds a given fixed time budget $\tau>0$. As such, the total number of pulls is random. We receive the reward of the $i$-th arm pull after this completion time, and can make a new decision if there is remaining time budget. We have clarified on this in the paper.
>
> 1.3 Thank you for your valuable remarks. We have modified the introduction parts to move the motivating examples before problem formulation and added a new section informally describing the main results.
>
> 1.4 We thank the reviewer for pointing this out. Our UCB-type algorithms do not require prior knowledge of higher-order moments (e.g., upper bounds on $(1+\epsilon)$-moments) as they utilize empirical Bernstein inequalities that employ empirical variance estimators. These empirical Bernstein inequalities require finite fourth-order moments (Minsker, 2015). As the reviewer pointed out, we could have instead assumed bounded $(1+\epsilon)$-moments for some $\epsilon\in(0,1]$, but in that case the UCB-type algorithms would require knowing upper bounds on these moments, and the sharpness of these bounds would depend on this prior knowledge. We avoid this by using empirical Bernstein inequalities. We have now clarified this point in the revision.
>
> 1.5 Intuitively, the resetting time is defined as the time between the interruption of an ongoing arm pull and the next arm pull. More concretely, note that the $n$-th pull of arm $k$ takes $X_{k,n}$ completion time. If this arm pull is interrupted at time $t$ before its completion (i.e., $t < X_{k,n}$), then it takes $t+C_{k,n}(t)$ time until the next arm pull, and $C_{k,n}(t)$ models the time until the process returns to its initial.
>
> 1.6 Intuitively, the reward rate characterize how the expected reward grows for a given action on the (arm, interrupt time) per unit time.  We can think about it as the average reward gain per unit time. For a simpler case without interruption, if the recurrent renewal process has reward and time cost$\{(R_{k,t},X_{k,t}):t\in\mathbb{N}\}$, then $\lim_{\tau\rightarrow\infty}\sum_{t<\tau}R_{k,t}/\sum_{t<\tau}X_{k,t}\rightarrow \frac{\mathbb{E}[R_{k,1}]}{\mathbb{E}[X_{k,1}]}$, which is the reward rate.
>
> 2. Thanks for pointing this out the limitation without lower bounds. We agree that ideally lower bounds should be developed to fully characterize the theoretical results. However, in this work, given the random stopping time, there is the technical challenge on finding the lower bound only using empirical estimations without higher moment knowledge. It would be a good future study to study.
>
> 3. Thanks for this suggestion. We have added the time per step data in the simulation section.
>
> 4. Thanks for suggesting these related works on delayed feedback bandits. We have added the discussions in the related work section. Specifically, we agree that similar to the delayed feedback bandits, our setup has a waiting time before the reward can be collected. However, the key difference here is that the time here is treated as a cost and there is the additional control on interrupting an arm. We have made this distinction clearer in the revision.
>
> 5. Thanks for suggesting these related works on stochastic bandits. We would like to clarify the difference between these restart mechanisms. For non-stochastic environments that the reviewer mentioned, the restarting is for the sampling process, for example, using a sliding window or discounted factor to forget obsolete information. However, in our setup, the restart mechanism implies interrupting an ongoing arm pull in order to achieve higher reward-per-unit-time under heavy-tailed completion time distributions. It does not mean restarting statistical estimation. Our work is more closely following the renewal theory framework. We have clarified on this in the revision.

---

### Review · Reviewer_GTM7 · 2025-03-29

**Summary Of Contributions:**

This work presents an extension to the multi-armed bandit (MAB) problem by integrating controlled restarts within a budgeted framework. Unlike traditional MAB settings that focus solely on arm selection to maximize rewards over a fixed number of rounds, this work considers a continuous-time scenario where each arm pull incurs a random completion time and yields a random, possibly correlated reward, both subject to a strict time budget $\tau$. The decision-maker can interrupt an ongoing task, forgoing its potential reward, to restart with a potentially more rewarding arm, incurring a deterministic resetting cost $C_k(t)$. The work is motivated with real-world applications such as algorithm portfolios for SAT solvers, task scheduling in distributed servers , and hyperparameter tuning in neural network training, where tasks exhibit stochastic durations and restarts can optimize performance. The authors propose three algorithms: UCB-RM and UCB-RB for finite restart time sets, achieving $O(\log \tau)$ regret, and UCB-RC for continuous restart time sets, achieving $O(\sqrt{\tau \log \tau})$ regret. These algorithms operate without prior knowledge of system statistics, leveraging right-censored feedback to inform decisions. Theoretical guarantees are substantiated with empirical evaluations with superior performance over baseline restart strategies like the Luby scheme.

**Audience:**

Yes

**Claims And Evidence:**

Yes

**Requested Changes:**

See the weaknesses, a brief justification would be appreciated.

Generally the theoretical framework for this work is robust, and the results well presented and well justified.

**Strengths And Weaknesses:**

Strengths
This is a well written paper, with well presented and sound theoretical results. The formulation is novel and allows for several benefits, like being versatile for diverse domains with a unified frameworkk. It also promises robustness to unknown statistics, which increases practical utility, unlike Cayci et al. (2019). Exploiting right-censored feedback also reduces regret dependence on $L$, a novel contribution over BwK.

Weaknesses
- Discretization: UCB-RC’s fixed $\delta$ may not adapt to local reward rate variations potentially leading to sub-optimal performance. Dynamic discretization could be beneficial.
- The assumption of deterministic and known resetting costs raise the question of how realistic it is. Stochastic or unknown costs would be common in practice - how would this affect the results?

---

> ### Author Response · Authors · 2025-05-20
> **Response to Reviewer GTM7**
>
> We would like to thank the reviewer for the very valuable and constructive feedback, which have helped us improve the overall quality of the paper. Below, we provide our responses to all questions and comments by the reviewer. We updated the original paper, with changes highlighted in blue.
>
> 1. Thanks for making this valuable suggestion on dynamically changing $\delta$. We agree that adapting the parameter $\delta$ dynamically may give to better performance. For this work, we selected fixed $\delta$ to simplify the analysis. It would definitely be an interesting future work to look into the analysis for adaptations on the discretization parameter.
>
> 2. Thanks for this valuable point on making the model more general. We agree that using stochastic or unknown resetting costs would make the setup more practical. Making that change would add difficulty in the analysis with additional random variable and empirical estimation in determining the bound of the control process. This would definitely be worth looking into in details as a potential extension of the current work.

---

### Review · Reviewer_ddV8 · 2025-05-13

**Summary Of Contributions:**

This work studied the bandit problem with controlled restarts. It proposed a few algorithms and derived the corresponding upper bounds on the reward rate. It also evaluated the algorithms with numerical experiments.

**Audience:**

Yes

**Claims And Evidence:**

Yes

**Requested Changes:**

Please refer to 'Weaknesses' in the part of *Strengths and Weaknesses*.

**Strengths And Weaknesses:**

**Strengths:**
1. The organization of the paper is good.
2. A few motivating scenarios are discussed.
3. Many relevant works are discussed.

**Weaknesses:** some more description and explanation are appreciated.
1. Why are the total time and reward defined as in equations (3) and (4)?
1. I am a bit confused by the dynamic. Does the k+1-th trial start only after the termination of the k-th trial?
1. How are the motivating examples described in Section 3 related to the formulated problem?
1. What is the key challenge of this work?
1. Why is the reward rate applied as the quantity to evaluate algorithms?
1. When will the agent know the statistics?
1. In numerical experiments, the benchmark algorithm is form a work published in 1993. Is there any more recent work/algorithm? How about those works mentioned in Section 1.1?

---

> ### Author Response · Authors · 2025-05-20
> **Response to Reviewer ddV8**
>
> We would like to thank the reviewer for the very valuable and constructive feedback, which have helped us improve the overall quality of the paper. Below, we provide our responses to all questions and comments by the reviewer. We updated the original paper, with changes highlighted in blue.
>
> 1. Thanks for asking for this clarification question on definitions. The reward and time cost are defined to capture the interruption dynamics. (a) if there is no interruption and the job finishes, we will get a reward with the cost of the actual time the job takes. (b) Otherwise, there will be no reward and the cost is the interruption time plus an additional restarting cost.
>
> 2. Thanks for inquiring on this details of the dynamics. A new trial can start in two cases: (a) the actual running time is less than the interruption time, so the controller will let the job finish; (b) the running time is larger than the interruption time, the controller will interrupt it for a potentially better one.
>
> 3. Thanks for asking for clarifications on motivations. The motivating examples all have the underlying structure that the controller needs to make decisions under a time budget and the possibility to restart. That is why we formulate the time cost and reward as in Equations (3) and (4).
>
> 4. The key challenge is that the system is under a random stopping time, and we do not assume any prior knowledge on higher moment statistics of the system. We carefully designed the control process and did the regret analysis using renewal theory, stochastic control and concentration inequalities for reward rate estimation.
>
> 5. The reward rate can be viewed as the growth rate of the total expected reward over time if the controller persistently chooses an action. We can view it as an time average reward per unit time. We plot the reward rate to show that the empirical distribution actually has a heavy tail distribution to justify for the interruption mechanism.
>
> 6. In this setup, the agent/controller does not assume to know the statistics of the system. Instead, it uses empirical estimation (median-based or empirical-mean based) to make the decisions.
>
> 7. Thanks for checking on numerical comparison. There have not been existing works that fit into the setup we have. The benchmark we chose is the most classic one whose performance can be manually tuned to be used as a good baseline.

---

### Comment · Action_Editor_2VKd · 2025-05-14
**Start of discussion period**

Dear Authors,

Now that three reviews are available. All three reviews have a fair number of questions. Two reviewers consider this paper to be generally well-written, while the other reviewer considers the presentation to be improved. Please read them carefully and answer the questions raised. The reviewers can submit a review in two weeks, and the goal during this period is to provide the information so that reviewers are comfortable submitting a decision recommendation.

Best,
AE

---

> ### Author Response · Authors · 2025-05-16
> **Starting the discussion**
>
> Dear AE and Reviewers,
>
> We thank you all for providing your reviews and remarks about our paper. We benefited greatly from the questions that are posed by the reviewers. We have already made updates and will continue to address the recent review in our paper and we are happy to interact with the reviewers within this allowed 2 week-period. We hope to send our responses with an update as soon as possible, and happy to receive further feedback from the reviewers.
>
> Thank you once again for all your service, which helped us greatly in improving the content and presentation of our work.
>
> Best,
> Authors

---

> > ### Comment · Action_Editor_2VKd · 2025-05-19
> >
> > Dear Authors,
> >
> > Thank you for your efforts in updating the paper. We are looking for your responses.
> >
> > Best,
> >
> > AE

---

> > > ### Author Response · Authors · 2025-05-20
> > > **Responses Posted**
> > >
> > > We would like to thank the reviewers for the very valuable and constructive feedback, which have helped us improve the overall quality of the paper. We have provided our responses to all questions and comments by the reviewer. Also, we updated the original paper, with changes highlighted in blue.
> > >
> > > Best,
> > > Authors

---

### Decision · Action_Editor_2VKd · 2025-06-24

**Recommendation:** Accept with minor revision

**Additional Comments:**

The paper studies a budgeted bandit problem with known costs $C_k(t)$, heavy-tailed completion time $X_{k,n}$ (bounded $4$-th moments), and bounded rewards $R_{k,n} \in [0,1]$. The algorithm can switch an arm when the completion time is too slow. The key quantity is $U_{k,n}(t)$ that indicates the completion time plus switching cost. The paper proposes algorithms based on upper confidence bounds (UCBs), which are popular in these problems. That said, the construction of UCB has many new insights, such as the use of a median-based estimator of $U_{k,n}(t)$ as well as discretization for continuous decision time. Numerical experiments verified the performance of the proposed learning algorithms for three different applications: algorithm portfolios for SAT solvers, task scheduling in distributed networks, and tuning the stopping time for neural network training.

All three reviewers are (weakly) positive. Reviewer ddV8 has many clarification questions that helped to revise the paper. Reviewer GTM7 considers the use of hyperparameter $\delta$ is a bit artificial, as well as fixed known resetting cost. Reviewer wmWz raised clarification questions as well as fundamental contribution such as the lack of lower bounds.

I also took a look at the paper, and it looks great. However, I have some questions regarding the problem setting. Given that we can proceed with further steps given these are clear, I recommend a minor revision.

Clarification is required for the following points:
*  For a given arm $k$, the joint variable $(X_{k,n}, R_{k,n})$ is sampled identically for every $n$, right?
* On the reward assumption: I assume the completion time $X_{k,n}$ as bounded moments for all $p = 1,2,3,...$ as (1) is for all $p \ge 4$ and 4-th order moment implies the lower order (or it may be the typo of $p \le 4$).
* Bernstein's inequality requires the boundedness of the random variable (true?), whereas this paper assumes no boundedness on $X_{k,n}$. Please clarify the concentration inequalities on $X_{k,n}$ that this paper is using.
* I guess $> t$ in Eq. (10) is misplaced (inside the 1 operator).
* I guess $r^*$ is usually achieved by taking $t \rightarrow \infty$. I mean, if we know the optimal arm, we never switch for a reasonable class of switching cost? In that sense, I guess no argmax exists, as any finite $t < \infty$ is slightly suboptimal (it is argsup?).
* I think an asymptotic lower bound is doable given an asymptotic performance upper bound is given ("OPT"), but I am fine if Reviewer wmWz is okay with it.

Best,

AE

**Audience:**

Yes

**Audience Explanation:**

Bandits with continuous-time decision making and variable costs are surely of significant interest in the online learning community. Most existing papers are not compatible with these criteria, whereas in many real-world problems requires these.

**Claims And Evidence:**

Yes

**Claims Explanation:**

The paper is a theoretical machine learning paper, and most of the discussion in the paper is fine. I need several clarifications before acceptance, and thus recommend a minor revision.

---

> ### Author Response · Authors · 2025-07-11
>
> We would like to thank the AE and the reviewers for the very valuable and constructive feedback, which have helped us improve the overall quality of the paper. We updated the revised paper, with changes highlighted in blue. We will upload the camera-ready version upon receiving feedback from the action editor.
>
> Below, we provide our responses to all questions and comments.
>
> 1. Thanks for asking for this clarification question. The joint variable is sampled identically for every $n$, we have made that point clearer in the problem formulation.
>
> 2. Thanks for inquiring on this details. It was a typo. The correct condition should be there exists $\exists p\geq 4$. We have made the change for equation (1).
>
> 3. In order to use Bernstein's inequality, we assume the set of restarting time to be bounded. Therefore, $X_{k,n}$ will be truncated and bounded. We have highlighted this in the summary of main results (Section 3).
>
> 4. Thanks for pointing out this typo. We have corrected in the revision.
>
> 5. Depending on the distribution of the completion time, $t$ can be finite or infinite. We have updated the argmax in equation (11) to argsup as suggested.
>
> 6. We agree with the reviewer that regret lower bounds would be very helpful in completely characterizing the theoretical results. However, in this work, the random time-horizon, which depends on the history of actions as well as random completion times, creates a significant technical challenge compared to the stochastic bandit problem. As such, we consider it as an important and interesting future work.

---

> > ### Comment · Action_Editor_2VKd · 2025-07-15
> >
> > Dear Authors,
> >
> > Thank you for your revision. The paper looks better! It would be great if you could quickly check the following.
> >
> > * argsup also applies to Algorithm 1 and (15) as well. I think the argsup always applies to the case where the fact that $X_{k,n} > t$ does not give you much information, such as an exponential distribution, which may not be your primary interest. Could you provide some insight when you choose action $(k, \infty)$? That is, to wait for an arbitrarily long completion time.
> > * "if the set of restarting time is bounded" -> I think any real finite set is bounded. Could you elaborate what you meant?
> >
> > Thanks!
> >
> > Best,
> >
> > AE

---

> ### Author Response · Authors · 2025-07-18
>
> We would like to thank the AE for the follow-up questions. Below we provide our answers. We also update the paper accordingly.
>
> - Thank you for pointing out the typos in Algorithm 1 and Equation (15). We have corrected them in the paper as you suggested. The case $(k^\star, \infty)$ is an optimal action emerges is fully characterized by Proposition 6 in terms of the residual reward rate. In case the completion time $X_{k^\star,n}$ and the reward $R_{k^\star,n}$ are independent random variables, $(k^\star, \infty)$ is an optimal action if the mean residual time $E[X_{k^\star,n}-t|X_{k^\star,n} > t]$ is a decreasing function of $t > 0$. Intuitively, this corresponds to completion time distributions with lighter tails compared to the exponential distribution, which is the boundary case.
>
> - We have updated Propositions 1 and 2. In that part, we clarify the exclusion of $\infty$ from the set of possible restart times, which implies that each arm pull will be restarted after a finite time. This enables the use of Bernstein-type upper confidence bounds with improved constants, as presented in Proposition 2.